# Optimal Preconditioning and Fisher Adaptive Langevin Sampling

**Michalis K. Titsias**
Google DeepMind
mtitsias@google.com

## Abstract

We define an optimal preconditioning for the Langevin diffusion by analytically optimizing the expected squared jumped distance. This yields as the optimal preconditioning an inverse Fisher information covariance matrix, where the covariance matrix is computed as the outer product of log target gradients averaged under the target. We apply this result to the Metropolis adjusted Langevin algorithm (MALA) and derive a computationally efficient adaptive MCMC scheme that learns the preconditioning from the history of gradients produced as the algorithm runs. We show in several experiments that the proposed algorithm is very robust in high dimensions and significantly outperforms other methods, including a closely related adaptive MALA scheme that learns the preconditioning with standard adaptive MCMC as well as the position-dependent Riemannian manifold MALA sampler.

## 1 Introduction

Markov chain Monte Carlo (MCMC) is a general framework for simulating from arbitrarily complex distributions, and it has shown to be useful for statistical inference in a wide range of problems [18, 11]. The main idea of an MCMC algorithm is quite simple. Given a complex target $\pi(x)$, a Markov chain is constructed using a $\pi$-invariant transition kernel that allows to simulate dependent realizations $x_1, x_2, \ldots$ that eventually converge to samples from $\pi$. These samples can be used for Monte Carlo integration by forming ergodic averages. A general way to define $\pi$-invariant transition kernels is the Metropolis-Hastings accept-reject mechanism in which the chain moves from state $x_n$ to the next state $x_{n+1}$ by first generating a candidate state $y_n$ from a proposal distribution $q(y_n|x_n)$ and then it sets $x_{n+1} = y_n$ with probability $\alpha(x_n, y_n)$:

$$\alpha(x_n, y_n) = \min(1, a_n), \;\; a_n = \frac{\pi(y_n)}{\pi(x_n)} \frac{q(x_n|y_n)}{q(y_n|x_n)}, \tag{1}$$

or otherwise rejects $y_n$ and sets $x_{n+1} = x_n$. The choice of the proposal distribution $q(y_n|x_n)$ is crucial because it determines the mixing of the chain, i.e. the dependence of samples across time. For example, a "slowly mixing" chain even after convergence may not be useful for Monte Carlo integration since it will output a highly dependent set of samples producing ergodic estimates of very high variance. Different ways of defining $q(y_n|x_n)$ lead to common algorithms such as random walk Metropolis (RWM), Metropolis-adjusted Langevin algorithm (MALA) [40, 35] and Hamiltonian Monte Carlo (HMC) [15, 28]. Within each class of these algorithms adaptation of parameters of the proposal distribution, such as a step size, is also important and this has been widely studied in the literature by producing optimal scaling results [34, 35, 36, 22, 6, 8, 38, 7, 39, 9], and also by developing adaptive MCMC algorithms [21, 5, 37, 19, 2, 1, 4, 24]. The standard adaptive MCMC procedure in [21] uses the history of the chain to recursively compute an empirical covariance of the target $\pi$ and build a multivariate Gaussian proposal distribution. However, this type of covariance adaptation can be too slow and not so robust in high dimensional settings [38, 3].

37th Conference on Neural Information Processing Systems (NeurIPS 2023).

In this paper, we derive a fast and very robust adaptive MCMC technique in high dimensions that learns a preconditioning matrix for the MALA method, which is the standard gradient-based MCMC algorithm obtained by a first-order discretization of the continuous-time Langevin diffusion. Our first contribution is to define an optimal preconditioning by analytically optimizing a criterion on the Langevin diffusion. The criterion is the well-known expected squared jumped distance [30] which at optimum yields as a preconditioner the inverse matrix $\mathcal{I}^{-1}$ of the following Fisher information covariance matrix $\mathcal{I} = \mathbb{E}_{\pi(x)}[\nabla \log \pi(x) \nabla \log \pi(x)^\top]$. This contradicts the common belief in adaptive MCMC that the covariance of $\pi$ is the best preconditioner. While this is a surprising result we show that $\mathcal{I}^{-1}$ connects with a certain quantity appearing in optimal scaling of RWM [34, 36].

Having recognized $\mathcal{I}^{-1}$ as the optimal preconditioning we derive an easy to implement and computationally efficient adaptive MCMC algorithm that learns from the history of gradients produced as MALA runs. This method sequentially updates an empirical inverse Fisher estimate $\hat{\mathcal{I}}_n^{-1}$ using a recursion having quadratic cost $O(d^2)$ ($d$ is the dimension of $x$) per iteration. In practice, since for sampling we need a square root matrix of $\hat{\mathcal{I}}_n^{-1}$ we implement the recursions over a square root matrix by adopting classical results from Kalman filtering [31, 10]. We compare our method against MALA that learns the preconditioning with standard adaptive MCMC [21], a position-dependent Riemannian manifold MALA [20] as well as simple MALA (without preconditioning) and HMC. In several experiments we show that the proposed algorithm significantly outperforms all other methods.

## 2 Background

We consider an intractable target distribution $\pi(x)$ with $x \in \mathbb{R}^d$, known up to some normalizing constant, and we assume that $\nabla \log \pi(x) := \nabla_x \log \pi(x)$ is well defined. A continuous time process with stationary distribution $\pi$ is the overdamped Langevin diffusion

$$dx_t = \frac{1}{2} A \nabla \log \pi(x_t) dt + \sqrt{A} dB_t, \tag{2}$$

where $B_t$ denotes $d$-dimensional Brownian motion. This is a stochastic differential equation (SDE) that generates sample paths such that for large $t$, $x_t \sim \pi$. We also incorporate a *preconditioning matrix* $A$, which is a symmetric positive definite covariance matrix, while $\sqrt{A}$ is such that $\sqrt{A}\sqrt{A}^\top = A$.

Simulating from the SDE in (2) is intractable and the standard approach is to use a first-order Euler-Maruyama discretization combined with a Metropolis-Hastings adjustment. This leads to the so called preconditioned *Metropolis-adjusted Langevin algorithm* (MALA) where at each iteration given the current state $x_n$ (where $n = 1, 2, \ldots$) we sample $y_n$ from the proposal distribution

$$q(y_n|x_n) = \mathcal{N}(y_n|x_n + \frac{\sigma^2}{2} A \nabla \log \pi(x_n), \sigma^2 A), \tag{3}$$

where the step size $\sigma^2 > 0$ appears due to time discretization. We accept $y_n$ with probability $\alpha(x_n, y_n) = \min(1, a_n)$ where $a_n$ follows the form in (1). The obvious way to compute $a_n$ is

$$a_n = \frac{\pi(y_n)}{\pi(x_n)} \frac{q(x_n|y_n)}{q(y_n|x_n)} = \frac{\pi(y_n)}{\pi(x_n)} \frac{\exp\{-\frac{1}{2\sigma^2}||x_n - y_n - \frac{\sigma^2}{2} A \nabla \log \pi(y_n)||^2_{A^{-1}}\}}{\exp\{-\frac{1}{2\sigma^2}||y_n - x_n - \frac{\sigma^2}{2} A \nabla \log \pi(x_n)||^2_{A^{-1}}\}},$$

where $||z||^2_{A^{-1}} = z^\top A^{-1} z$. However, in some cases that involve high dimensional targets, this can be costly since in the ratio of proposal densities both the preconditioning matrix $A$ and its inverse $A^{-1}$ appear. In turns out that we can avoid $A^{-1}$ and simplify the computation as stated below.

**Proposition 1.** *For preconditioned MALA with proposal density given by* (3) *the ratio of proposals in the M-H acceptance probability can be written as*

$$\frac{q(x_n|y_n)}{q(y_n|x_n)} = \exp\{h(x_n, y_n) - h(y_n, x_n)\}, \quad h(z, v) = \frac{1}{2}\left(z - v - \frac{\sigma^2}{4} A \nabla \log \pi(v)\right)^\top \nabla \log \pi(v).$$

This expression does not depend on the inverse $A^{-1}$, and this leads to computational gains and simplified implementation that we exploit in the adaptive MCMC algorithm presented in Section 4.

The motivation behind the use of preconditioned MALA is that with a suitable preconditioner $A$ the mixing of the chain can be drastically improved, especially for very anisotropic target distributions.

A very general way to specify $A$ is by applying an adaptive MCMC algorithm, which learns $A$ online. To design such an algorithm it is useful to first specify a notion of optimality. A common argument in the literature, that is used for both RWM and MALA, is that a suitable $A$ is the unknown covariance matrix $\Sigma$ [21, 38, 24] of the target $\pi$. This means that we should learn $A$ so that to approximate $\Sigma$. However, this argument is rather heuristic since it is not based on an optimality criterion. One of our contributions is to specify an optimal $A^*$ based on an optimization procedure, that we describe in Section 3. This $A^*$ will turn out to be not the covariance matrix of the target but an inverse Fisher information matrix.

## 3    Optimal preconditioning using expected squared jumped distance

Preconditioning aims to improve sampling when different directions (or individual variables $x_i$) in the state space can have different scalings under the target $\pi$. Here, we develop a method for selecting the preconditioning through the optimization of an objective function. This method uses the observation that an effective preconditioning correlates with large values of the global step size $\sigma^2$ in MALA, i.e. $\sigma^2$ is allowed to increase when preconditioning becomes effective as shown in the sampling efficiency scores in Table 1 and the corresponding estimated step sizes reported in Appendix E.1.

In our analysis we consider the rejection-free or unadjusted Langevin sampler where we discretize the time continuous Langevin diffusion in (2) with a small finite $\delta := \sigma^2 > 0$ so that

$$x_{t+\delta} - x_t = \frac{\delta}{2} A \nabla \log \pi(x_t) + \sqrt{A}(B_{t+\delta} - B_t), \text{ where } B_{t+\delta} - B_t \sim \mathcal{N}(0, \delta I). \quad (4)$$

We will use the expected squared jumped distance $J(\delta, A) = \mathbb{E}[||x_{t+\delta} - x_t||^2]$ computed as follows.

**Proposition 2.** *If $x_t \sim \pi(x_t)$ the vector $x_{t+\delta} - x_t$ defined by (4) has zero mean and covariance*

$$\mathbb{E}[(x_{t+\delta} - x_t)(x_{t+\delta} - x_t)^\top] = \frac{\delta^2}{4} A \mathbb{E}_{\pi(x_t)} \left[ \nabla \log \pi(x_t) \nabla \log \pi(x_t)^\top \right] A + \delta A. \quad (5)$$

*Further, $tr \left( \mathbb{E}[(x_{t+\delta} - x_t)(x_{t+\delta} - x_t)^\top] \right) = \mathbb{E}[tr \left( (x_{t+\delta} - x_t)(x_{t+\delta} - x_t)^\top \right)] = \mathbb{E}[||x_{t+\delta} - x_t||^2]$, which shows that $J(\delta, A)$ is the trace of the covariance matrix in (5).*

To control discretization error we impose an upper bound constraint $J(\delta, A) \leq \epsilon$ for a small $\epsilon > 0$. A preconditioning that "symmetrizes" the target can be obtained by maximizing the discretization step size $\delta$ subject to $J(\delta, A) \leq \epsilon$. Since $J(\delta, A)$ monotonically increases with $\delta$, the maximum $\delta^*$ satisfies $\min_A J(\delta^*, A) = \epsilon$. This means that the *optimal* preconditioning $A^*$ is obtained by minimizing $J(\delta, A)$ under some global scale constraint on $A$, as stated next.

**Proposition 3.** *Suppose $A$ is a symmetric positive definite matrix satisfying $tr(A) = c$, with $c > 0$ a constant. Then the objective $J(\delta, A)$, for any $\delta > 0$, is miminized for $A^*$ given by*

$$A^* = k\mathcal{I}^{-1}, \quad k = \frac{c}{\sum_{i=1}^{d} \frac{1}{\mu_i}}, \quad \mathcal{I} = \mathbb{E}_{\pi(x)} \left[ \nabla \log \pi(x) \nabla \log \pi(x)^\top \right], \quad (6)$$

*where $\mu_i$s are the eigenvalues of $\mathcal{I}$ assumed to satisfy $0 < \mu_i < \infty$.*

The positive multiplicative scalar $k$ in (6) is not important since the specific value $c > 0$ is arbitrary, e.g. if we choose $c = \sum_{i=1}^{d} \frac{1}{\mu_i}$ then $k = 1$ and $A^* = \mathcal{I}^{-1}$. In other words, what matters is that the optimal $A^*$ is proportional to the inverse matrix $\mathcal{I}^{-1}$, so it follows the curvature of $\mathcal{I}^{-1}$. For a multivariate Gaussian $\pi(x) = \mathcal{N}(x|\mu, \Sigma)$ it holds $\mathcal{I}^{-1} = \Sigma$, so the optimal preconditioner coincides with the covariance matrix of $x$. More generally though, for non-Gaussian targets this will not hold.

**Connection with classical Fisher information matrix.**    The matrix $\mathcal{I}$ is positive definite since it is the covariance of the gradient $\nabla \log \pi(x) := \nabla_x \log \pi(x)$ where $\mathbb{E}_{\pi(x)}[\nabla \log \pi(x)] = 0$. Also, $\mathcal{I}$ is similar to the classical Fisher information matrix. To illustrate some differences suppose that the target $\pi(x)$ is a Bayesian posterior $\pi(\theta|Y) \propto p(Y|\theta)p(\theta) = p(Y, \theta)$ where $Y$ are the observations and $\theta := x$ are the random parameters. The classical Fisher information is a *frequentist* quantity where we fix some parameters $\theta$ and compute $G(\theta) = \mathbb{E}_{p(Y|\theta)}[\nabla_\theta \log p(Y|\theta) \nabla_\theta \log p(Y|\theta)^\top]$ by averaging over data. In contrast, $\mathcal{I} = \mathbb{E}_{p(\theta|Y)}[\nabla_\theta \log p(Y, \theta) \nabla_\theta \log p(Y, \theta)^\top]$ is more like a *Bayesian* quantity where we fix the data $Y$ and average over the parameters $\theta$. Importantly, $\mathcal{I}$ is not a function of $\theta$ while $G(\theta)$ is. Similarly to the classical Fisher information, $\mathcal{I}$ also satisfies the following standard property: Given that $\log \pi(x)$ is twice differentiable and $\nabla_x^2 \log \pi(x)$ is the Hessian matrix, $\mathcal{I}$ from (6) is also written as $\mathcal{I} = -\mathbb{E}_{\pi(x)}[\nabla_x^2 \log \pi(x)]$. Next, we refer to $\mathcal{I}$ as the Fisher matrix.

**Connection with optimal scaling.** The Fisher matrix $\mathcal{I}$ connects also with the optimal scaling result for the RWM algorithm [34, 36]. Specifically, for targets of the form $\pi(x) = \prod_{i=1}^d f(x_i)$, the RWM proposal $q(y_n|x_n) = \mathcal{N}(y_n|x_n, (\sigma^2/d)I_d)$ and as $d \to \infty$ the optimal parameter $\sigma^2$ is $\sigma^2 = \frac{2.38}{J}$ where $J = \mathbb{E}_{f(x)}[(\frac{d\log f(x)}{dx})^2]$ is the (univariate) Fisher information for the univariate density $f(x)$, and the preconditioning involves as in our case the inverse Fisher $\mathcal{I}^{-1} = \frac{1}{J}I_d$. This result has been generalized also for heterogeneous targets in [36] where again the inverse Fisher information matrix (having now a more general diagonal form) appears as the optimal preconditioner.

## 4 Fisher information adaptive MALA

Armed with the previous optimality result, we wish to develop an adaptive MCMC algorithm to optimize the proposal in (3) by learning online the global variance $\sigma^2$ and the preconditioner $A$. For $\sigma^2$ we follow the standard practice to tune this parameter in order to reach an average acceptance rate around $0.574$ as suggested by optimal scaling results [35, 36]. For the matrix $A$ we want to adapt it so that approximately it becomes proportional to the inverse Fisher $\mathcal{I}^{-1}$ from (6). We also incorporate a parametrization that helps the adaptation of $\sigma^2$ to be more independent from the one of $A$. Specifically, we remove the global scale from $A$ by defining the overall proposal as

$$q(y_n|x_n) = \mathcal{N}\left(y_n|x_n + \frac{\sigma^2}{\frac{2}{d}\mathrm{tr}(A)}A\nabla\log\pi(x_n), \frac{\sigma^2}{\frac{1}{d}\mathrm{tr}(A)}A\right), \tag{7}$$

where $\sigma^2$ is normalized by $\frac{1}{d}\mathrm{tr}(A)$, i.e. the average eigenvalue of $A$. Another way to view this is that the effective preconditioner is $A/(\frac{1}{d}\mathrm{tr}(A))$ which has an average eigenvalue equal to one. The proposal in (7) is invariant to any scaling of $A$, i.e. if $A$ is replaced by $kA$ (with $k > 0$) the proposal remains the same. Also, note that when $A$ is the identity matrix $I_d$ (or a multiple of identity) then $\frac{1}{d}\mathrm{tr}(I_d) = 1$ and the above proposal reduces to standard MALA with isotropic step size $\sigma^2$.

It is straightforward to adapt $\sigma^2$ towards an average acceptance rate $0.574$; see pseudocode in Algorithm 1. Thus our main focus next is to describe the learning update for $A$, in fact eventually not for $A$ itself but for a square root matrix $\sqrt{A}$ which is what we need to sample from the proposal in (7).

To start with, let us simplify notation by writing the score function at the $n$-th MCMC iteration as $s_n := \nabla_{x_n}\log\pi(x_n)$. We introduce the $n$-sample empirical Fisher estimate

$$\hat{\mathcal{I}}_n = \frac{1}{n}\sum_{i=1}^n s_i s_i^\top + \frac{\lambda}{n}I_d, \tag{8}$$

where $\lambda > 0$ is a fixed damping parameter. Given that certain conditions apply [21, 38] so that the chain converges and ergodic averages converge to exact expected values, $\hat{\mathcal{I}}_n$ is a consistent estimator satisfying $\lim_{n\to\infty}\hat{\mathcal{I}}_n = \mathcal{I}$ since as $n \to \infty$ the damping part $\frac{\lambda}{n}I_d$ vanishes. Including the damping is very important since it offers a Tikhonov-like regularization, similar to ridge regression, and it ensures that for any finite $n$ the eigenvalues of $\hat{\mathcal{I}}_n$ are strictly positive. An estimate then for the preconditioner $A_n$ can be set to be proportional to the inverse of the empirical Fisher $\hat{\mathcal{I}}_n$, i.e.

$$A_n \propto \left(\frac{1}{n}\sum_{i=1}^n s_i s_i^\top + \frac{\lambda}{n}I_d\right)^{-1} = n\left(\sum_{i=1}^n s_i s_i^\top + \lambda I_d\right)^{-1}. \tag{9}$$

Since any positive multiplicative scalar in front of $A_n$ plays no role, we can ignore the scalar $n$ and define $A_n = (\sum_{i=1}^n s_i s_i^\top + \lambda I_d)^{-1}$. Then, as MCMC iterates we can adapt $A_n$ in $O(d^2)$ cost per iteration based on the recursion

$$\text{Initialization: } A_1 = \left(s_1 s_1^\top + \lambda I_d\right)^{-1} = \frac{1}{\lambda}\left(I_d - \frac{s_1 s_1^\top}{\lambda + s_1^\top s_1}\right), \tag{10}$$

$$\text{Iteration: } A_n = \left(A_{n-1}^{-1} + s_n s_n^\top\right)^{-1} = A_{n-1} - \frac{A_{n-1} s_n s_n^\top A_{n-1}}{1 + s_n^\top A_{n-1} s_n}, \tag{11}$$

where we applied Woodbury matrix identity. This estimation in the limit can give the optimal preconditioning in the sense that under the ergodicity assumption, $\lim_{n\to\infty}\frac{A_n}{\mathrm{tr}(A_n)} = \frac{\mathcal{I}^{-1}}{\mathrm{tr}(\mathcal{I}^{-1})}$. In

practice we do not need to compute directly the matrix $A_n$ but a square root matrix $R_n := \sqrt{A_n}$, such that $R_n R_n^\top = A_n$, since we need a square root matrix to draw samples from the proposal in (7). To express the corresponding recursion for $R_n$ we will rely on a technique that dates back to the early days of Kalman filtering [31, 10], which applied to our case gives the following result.

**Proposition 4.** *A square root matrix $R_n$, such that $R_n R_n^\top = A_n$, can be computed recursively in $O(d^2)$ time per iteration as follows:*

$$\text{Initialization: } R_1 = \frac{1}{\sqrt{\lambda}} \left( I_d - r_1 \frac{s_1 s_1^\top}{\lambda + s_1^\top s_1} \right), \quad r_1 = \frac{1}{1 + \sqrt{\frac{\lambda}{\lambda + s_1^\top s_1}}} \tag{12}$$

$$\text{Iteration: } R_n = R_{n-1} - r_n \frac{(R_{n-1}\phi_n)\phi_n^\top}{1 + \phi_n^\top \phi_n}, \quad \phi_n = R_{n-1}^\top s_n, \quad r_n = \frac{1}{1 + \sqrt{\frac{1}{1 + \phi_n^\top \phi_n}}}. \tag{13}$$

A way to generalize the above recursive estimation of a square root for the inverse Fisher matrix is to consider the stochastic approximation framework [33]. This requires to write an online learning update for the empirical Fisher of the form

$$\hat{\mathcal{I}}_n = \hat{\mathcal{I}}_{n-1} + \gamma_n (s_n s_n^\top - \hat{\mathcal{I}}_{n-1}), \quad \text{initialized at } \hat{\mathcal{I}}_1 = s_1 s_1^\top + \lambda I_d, \tag{14}$$

where the learning rates $\gamma_n$ satisfy the standard conditions $\sum_{n=1}^{\infty} \gamma_n = \infty$, $\sum_{n=1}^{\infty} \gamma_n^2 < \infty$. Then, it is straightforward to generalize the recursion for the square root matrix in Proposition 4 to account for this more general case; see Appendix B. The recursion in Proposition 4 is a special case when $\gamma_n = \frac{1}{n}$. In our simulations we did not observe significant improvement by using more general learning rate sequences, and therefore in all our experiments in Section 5 we use the *standard* learning rate $\gamma_n = \frac{1}{n}$. Note that this learning rate is also used by other adaptive MCMC methods [21].

An adaptive algorithm that learns online from the score function vectors $s_n$ can work well in some cases, but still it can be unstable in general. One reason is that $s_n = \nabla \log \pi(x_n)$ will not have zero expectation when the chain is transient and states $x_n$ are not yet draws from the stationary distribution $\pi$. To analyze this, note that the learning signal $s_n$ enters in the empirical Fisher estimator $\hat{\mathcal{I}}_n$ through the outer product $s_n s_n^\top$ as shown by Eqs. (8) and (14). However, in the transient phase $s_n s_n^\top$ will be biased since the expectation $\mathbb{E}[s_n s_n^\top] = \mathbb{E}[(s_n - \mathbb{E}[s_n])(s_n - \mathbb{E}[s_n])^\top] + \mathbb{E}[s_n]\mathbb{E}[s_n]^\top \neq \mathcal{I}$, where the expectations are taken under the marginal distribution of the chain at time $n$. In practice the mean vector $\mathbb{E}[s_n]$ can take large absolute values, which can introduce significant bias through the additive term $\mathbb{E}[s_n]\mathbb{E}[s_n]^\top$. Thus, to reduce some bias we could track the empirical mean $\bar{s}_n = \frac{1}{n}\sum_{i=1}^{n} s_i$ and center the signal $s_n - \bar{s}_n$ so that the Fisher matrix is estimated by the empirical covariance $\frac{1}{n-1}\sum_{i=1}^{n}(s_i - \bar{s}_n)(s_i - \bar{s}_n)^\top$. The recursive estimation becomes similar to standard adaptive MCMC [21] where we recursively propagate an online empirical estimate for the mean of $s_n$ and incorporate it into the online empirical estimate of the covariance matrix (in our case the inverse Fisher matrix); see Eq. (17) in Section 5 for the standard adaptive MCMC recursion [21] and Appendix C for our Fisher method. While this can make learning quite stable we experimentally discovered that there is another scheme, presented next in Section 4.1, that is significantly better and stable especially for very anisotropic high dimensional targets; see detailed results in Appendix E.3.

## 4.1 Adapting to score function increments

An MCMC algorithm updates at each iteration its state according to $x_{n+1} = x_n + \mathbf{I}(u_n < \alpha(x_n, y_n))(y_n - x_n)$ where $\alpha(x_n, y_n)$ is the M-H probability, $u_n \sim U(0,1)$ is an uniform random number and $\mathbf{I}(\cdot)$ is the indicator function. This sets $x_{n+1}$ to either the proposal $y_n$ or the previous state $x_n$ based on the binary value $\mathbf{I}(u_n < \alpha(x_n, y_n))$. Similarly, we can consider the update of the score function $s(x) = \nabla \log \pi(x)$ and conveniently re-arrange it as an increment,

$$s_n^\delta = s(x_{n+1}) - s(x_n) = \mathbf{I}(u_n < \alpha(x_n, y_n))(s(y_n) - s(x_n)). \tag{15}$$

While both $s_n$ and $s_n^\delta$ have zero expectation when $x_n$ is from stationarity, i.e. $x_n \sim \pi$, the increment $s_n^\delta$ (unlike $s_n$) tends in practice to be more centered and close to zero even when the chain is transient, e.g. note that $s_n^\delta$ is zero when $y_n$ is rejected. Further, since the difference $s_n^\delta = s(x_{n+1}) - s(x_n)$ conveys information about the covariance of the score function we can use it in the recursion of

---
**Algorithm 1** Fisher adaptive MALA (blue lines are ommitted when not adapting $(R, \sigma^2)$)
---
**Input:** Log target $\log \pi(x)$; gradient $\nabla \log \pi(x)$; $\lambda > 0$ (default $\lambda = 10$); $\alpha_* = 0.574$
Initialize $x_1$ and $\sigma^2$ by running simple MALA (i.e. with $\mathcal{N}(y|x + (\sigma^2/2)\nabla \log \pi(x), \sigma^2 I)$) for $n_0$
(default 500) iterations where $\sigma^2$ is adapted towards acceptance rate $\alpha_*$
Initialize square root matrix $R = I_d$ and compute $(\log \pi(x_1), \nabla \log \pi(x_1))$
Initialize $\sigma_R^2 = \sigma^2$         *# placeholder for the normalized step size* $\sigma^2 / \frac{1}{d} tr(RR^\top)$
**for** For $n = 1, 2, 3, \ldots,$ **do**
   : Propose $y_n = x_n + (\sigma_R^2/2)R(R^\top \nabla \log \pi(x_n)) + \sigma_R R\eta, \ \ \eta \sim \mathcal{N}(0, I_d)$
   : Compute $(\log \pi(y_n), \nabla \log \pi(y_n))$
   : Compute $\alpha(x_n, y_n) = \min\left(1, e^{\log \pi(y_n) + h(x_n, y_n) - \log \pi(x_n) - h(y_n, x_n)}\right)$ by using Proposition 1
   : Compute adaptation signal $s_n^\delta = \sqrt{\alpha(x_n, y_n)}(\nabla \log \pi(y_n) - \nabla \log \pi(x_n))$
   : Use $s_n^\delta$ to adapt $R$ based on Proposition 4 (if $n = 1$ use (12) and if $n > 1$ use (13))
   : Adapt step size $\sigma^2 \leftarrow \sigma^2 \left[1 + \rho_n(\alpha(x_n, y_n) - \alpha_*)\right]$ *# default const learning rate* $\rho_n = 0.015$
   : Normalize step size $\sigma_R^2 = \sigma^2 / \frac{1}{d} tr(RR^\top)$        *#* $tr(RR^\top) = sum(R \circ R)$ *which is* $O(d^2)$
   : Accept/reject $y_n$ with probability $\alpha(x_n, y_n)$ to obtain $(x_{n+1}, \log \pi(x_{n+1}), \nabla \log \pi(x_{n+1}))$
**end for**
---

Proposition 4 to learn the preconditioner $A$, where we simply replace $s_n$ by $s_n^\delta$. As shown in the experiments this leads to a remarkably fast and effective adaptation of the inverse Fisher matrix $\mathcal{I}^{-1}$ without observable bias, or at least no observable for Gaussian targets where the true $\mathcal{I}^{-1}$ is known. We can further apply Rao-Blackwellization to reduce some variance of $s_n^\delta$. Since $s_n^\delta$ enters into the estimation of the empirical Fisher, see Eq. (8) or (14), through the outer product $s_n^\delta (s_n^\delta)^\top = \mathbf{I}(u_n < \alpha(x_n, y_n))(s(y_n) - s(x_n))(s(y_n) - s(x_n))^\top$ we can marginalize out the r.v. $u_n$ which yields $\mathbb{E}_{u_n}[s_n^\delta (s_n^\delta)^\top] = \alpha(x_n, y_n)(s(y_n) - s(x_n))(s(y_n) - s(x_n))^\top$. After this Rao-Blackwellization an alternative vector to use for adaptation is

$$s_n^\delta = \sqrt{\alpha(x_n, y_n)}(s(y_n) - s(x_n)), \tag{16}$$

which depends on the square root $\sqrt{\alpha(x_n, y_n)}$ of the M-H probability. As long as $\alpha(x_n, y_n) > 0$, the learning signal in (16) depends on the proposed sample $y_n$ even when it is rejected.

Finally, we can express the full algorithm for Fisher information adaptive MALA as outlined by Algorithm 1, which adapts by using the Rao-Blackwellized score function increments from Eq. (16). Note that, while Algorithm 1 uses $s_n^\delta$ from Eq. (16), the initial signal from Eq. (15) works equally well; see Appendix E.3. Also, the algorithm includes an initialization phase where simple MALA runs for few iterations to move away from the initial state, as discussed further in Section 5.

## 5 Experiments

### 5.1 Methods and experimental setup

We apply the Fisher information adaptive MALA algorithm (FisherMALA) to high dimensional problems and we compare it with the following other samplers. **(i)** The simple MALA sampler with proposal $\mathcal{N}(y_n|x_n + (\sigma^2/2)\nabla \log \pi(x_n), \sigma^2 I)$, which adapts only a step size $\sigma^2$ without having a preconditioner. **(ii)** A preconditioned adaptive MALA (AdaMALA) where the proposal follows exactly the from in (7) but where the preconditioning matrix is learned using standard adaptive MCMC based on the well-known recursion from [21]:

$$\mu_n = \frac{n-1}{n}\mu_{n-1} + \frac{1}{n}x_n, \ \Sigma_n = \frac{n-2}{n-1}\Sigma_{n-1} + \frac{1}{n}(x_n - \mu_{n-1})(x_n - \mu_{n-1})^\top, \tag{17}$$

where the recursion is initialized at $\mu_1 = x_1$ and $\Sigma_2 = \frac{1}{2}(x_2 - \mu_1)(x_2 - \mu_1)^\top + \lambda I$, and $\lambda > 0$ is the damping parameter that plays the same role as in FisherMALA. **(iii)** The Riemannian manifold MALA (mMALA) [20] which uses position-dependent preconditioning matrix $A(x)$. mMALA in high dimensions runs slower than other schemes since the computation of $A(x)$ may involve second derivatives and requires matrix decomposition that costs $O(d^3)$ per iteration. **(iv)** Finally, we include in the comparison the Hamiltonian Monte Carlo (HMC) sampler with a fixed number of 10 leap frog steps and identity mass matrix. We leave the possibility to learn with our method a preconditioner in HMC for future work since this is more involved; see discussion at Section 7.

For all experiments and samplers we consider $2 \times 10^4$ burn-in iterations and $2 \times 10^4$ iterations for collecting samples. We set $\lambda = 10$ in FisherMALA and AdaMALA. Adaptation of the proposal distributions, i.e. the parameter $\sigma^2$, the preconditioning or the step size of HMC, occurs only during burn-in and at collection of samples stage the proposal parameters are kept fixed. For all three MALA schemes the global step size $\sigma^2$ is adapted to achieve an acceptance rate around 0.574 (see Algorithm 1) while the corresponding parameter for HMC is adapted towards 0.651 rate [9]. In FisherMALA from the $2 \times 10^4$ burn-in iterations the first 500 iterations are used as the initialization phase in Algorithm 1 where samples are generated by just MALA with adaptable $\sigma^2$. Thus, only the last $1.95 \times 10^4$ burn-in iterations are used to adapt the preconditioner. For AdaMALA this initialization scheme proved to be unstable and we used a more elaborate scheme, as described in Appendix D.

We compute effective sample size (ESS) scores for each method by using the $2 \times 10^4$ samples from the collection phase. We estimate ESS across each dimension of the state vector $x$, and we report maximum, median and minimum values, by using the built-in method in TensorFlow Probability Python package. Also, we show visualizations that indicate sampling efficiency or effectiveness in estimating the preconditioner (when the ground truth preconditioner is known).

## 5.2 Gaussian targets

We consider three examples of multivariate Gaussian targets of the form $\pi(x) = \mathcal{N}(x|\mu, \Sigma)$, where the optimal preconditioner (up to any positive scaling) is the covariance matrix $\Sigma$ since the inverse Fisher is $\mathcal{I}^{-1} = \Sigma$. For such case the Riemannian manifold sampler mMALA [20] is the optimal MALA sampler since it uses precisely $\Sigma$ as the preconditioning. In contrast to mMALA which somehow has access to the ground-truth oracle, both FisherMALA and AdaMALA use adaptive recursive estimates of the preconditioner that should converge to the optimal $\Sigma$, and thus the question is which of them learns faster. To quantify this we compute the Frobenius norm $||\widetilde{A}_n - \widetilde{\Sigma}||_F$ across adaptation iterations $n$, where $\widetilde{B}$ denotes the matrix normalized by the average trace, i.e. $\widetilde{B} = B/(\frac{\text{tr}(B)}{d})$, for either $A_n$ given by FisherMALA or $A_n := \Sigma_n$ given by AdaMALA and where $\widetilde{\Sigma}$ is the optimal normalized preconditioner. The faster the Frobenius norm goes to zero the more effective is the corresponding adaptive scheme. For all three Gaussian targets the mean vector $\mu$ was taken to be the vector of ones and samplers were initialized by drawing from standard normal. The first example is a two-dimensional Gaussian target with covariance matrix $\Sigma = [1\ 0.995; 0.995\ 1]$. Both FisherMALA and AdaMALA perform almost the same (FisherMALA has faster convergence) in this low dimensional example as shown by Frobenius norm in Figure 1a; see also Figure 5 in the Appendix for visualizations of the adapted preconditioners. The following two examples involve 100-dimensional targets.

**Gaussian process correlated target.** We consider a Gaussian process to construct a 100-dimensional Gaussian where the covariance matrix is obtained by a non-stationary covariance function comprising the product of linear and squared exponential kernels plus small white noise, i.e. $[\Sigma]_{i,j} = s_i s_j \exp\{-\frac{1}{2}\frac{(s_i-s_j)^2}{0.09}\} + 0.001\delta_{i,j}$ where the scalar inputs $s_i$ form a regular grid in the range $[1, 2]$. Figure 1b shows the evolution of the Frobenius norms and panels d,c depict as $100 \times 100$ images the true covariance matrix and the preconditioner estimated by FisherMALA. For AdaMALA see Figure 6 in the Appendix. Clearly, FisherMALA learns much faster and achieves more accurate estimates of the optimal preconditioner. Further Table 1 shows that FisherMALA achieves significantly better ESS than AdaMALA and reaches the same performance with mMALA.

**Inhomogeneous Gaussian target.** In the last example we follow [28, 39] and we consider a Gaussian target with diagonal covariance matrix $\Sigma = \text{diag}(\sigma_1^2, \ldots, \sigma_{100}^2)$ where the standard deviation values $\sigma_i$ take values in the grid $\{0.01, 0.02, \ldots, 1\}$. This target is challenging because the different scaling across dimensions means that samplers with a single step size, i.e. without preconditioning, will adapt to the smallest dimension $x_1$ of the state while the chain at the higher dimensions, such as $x_{100}$, will be moving slowly exhibiting high autocorrelation. Note that FisherMALA and AdaMALA run without knowing that the optimal preconditioner is a diagonal matrix, i.e. they learn a full covariance matrix. Figure 2a shows the ESS scores for all 100 dimensions of $x$ for four samplers (except mMALA which has the same performance with FisherMALA), where we can observe that only FisherMALA is able to achieve high ESS uniformly well across all dimensions. In contrast, MALA and HMC that use a single step size cannot achieve high sampling efficiency and their

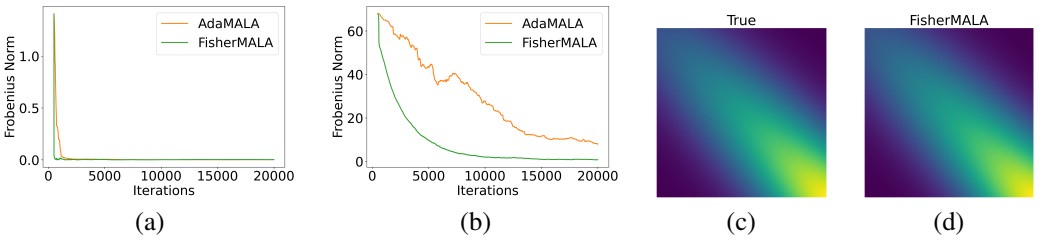

(a)  (b)  (c)  (d)

Figure 1: Panel (a) shows the Frobenius norm across burn-in iterations for the 2-D Gaussian and (b) for the GP target. The exact GP covariance matrix is shown in (c) and the estimated one by FisherMALA in (d).

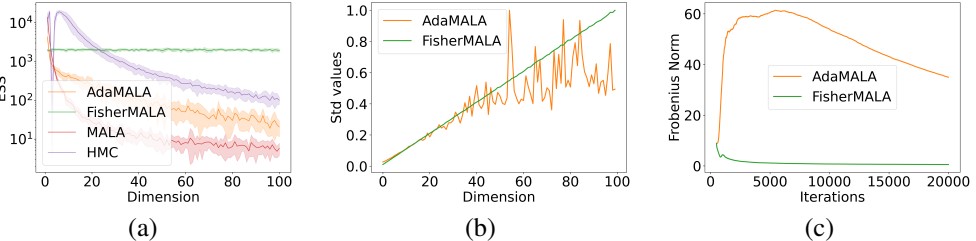

(a)  (b)  (c)

Figure 2: Results in the inhomogeneous Gaussian target.

ESS for dimensions close to $x_{100}$ drops significantly. The same holds for AdaMALA due to its inability to learn fast the preconditioner, as shown by the Frobenius norm values in Figure 2c and the estimated standard deviations in Figure 2b. AdaMALA can eventually get very close to the optimal precondtioner but it requires hundred of thousands of adaptive steps, while FisherMALA learns it with only few thousand steps.

### 5.3 Bayesian logistic regression

We consider Bayesian logistic regression distributions of the form $\pi(\theta|Y, Z) \propto p(Y|\theta, Z)p(\theta)$ with data $(Y, Z) = \{y_i, z_i\}_{i=1}^m$, where $z_i \in \mathbb{R}^d$ is the input vector and $y_i$ the binary label. The likelihood is $p(Y|\theta, Z) = \prod_{i=1}^m \sigma(\theta^\top z_i)^{y_i}(1 - \sigma(\theta^\top z_i))^{1-y_i}$, where $\theta \in \mathbb{R}^d$ are the random parameters assigned the prior $p(\theta) = \mathcal{N}(\theta|0, I_d)$. We consider six binary classification datasets (Australian Credit, Heart, Pima Indian, Ripley, German Credit and Caravan) with a number of data ranging from $n = 250$ to $n = 5822$ and dimensionality of the $\theta$ ranging from 3 to 87. We also consider a much higher 785-dimensional example on MNIST for classifying the digits 5 and 6, that has 11339 training examples. To make the inference problems more challenging, in the first six examples we do not standardize the inputs $z_i$ which creates very anisotropic posteriors over $\theta$. For the MNIST data, which initially are grey-scale images in [0, 255], we simply divide by the maximum pixel value, i.e. 255, to bring the images in [0, 1]. In Table 1 we report the ESS for the low 7-dimensional Pima Indians dataset, the medium 87-dimensional Caravan dataset and the higher 785-dimensional MNIST dataset, while the results for the remaining datasets are shown in Appendix E. Further, Figure 3 shows the evolution of the unnormalized log target density $\log\{p(Y|\theta, Z)p(\theta)\}$ for the best four samplers in

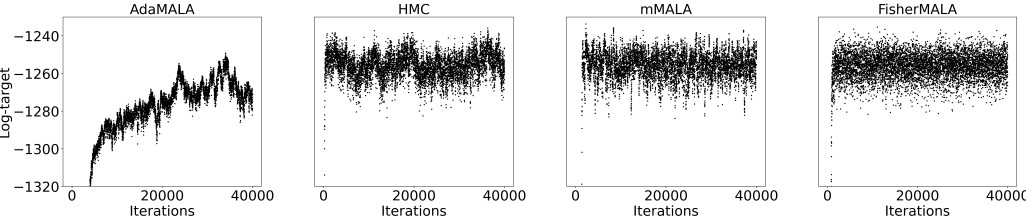

Figure 3: The evolution of the log-target across iterations for the best four algorithms in Caravan dataset.

Table 1: ESS scores are averages after repeating the simulations 10 times under different random initializations.

| | Max ESS | Median ESS | Min ESS |
|---|---|---|---|
| *GP target* ($d = 100$) | | | |
| MALA | $15.235 \pm 4.246$ | $6.326 \pm 1.934$ | $3.619 \pm 0.956$ |
| AdaMALA | $845.100 \pm 80.472$ | $662.978 \pm 81.127$ | $552.377 \pm 74.441$ |
| HMC | $17.625 \pm 6.706$ | $6.680 \pm 2.205$ | $4.315 \pm 0.889$ |
| mMALA | $2109.441 \pm 101.553$ | $2007.640 \pm 104.867$ | $1841.978 \pm 114.266$ |
| FisherMALA | $2096.259 \pm 94.751$ | $1923.753 \pm 95.820$ | $1784.962 \pm 104.440$ |
| *Pima Indian* ($d = 7$) | | | |
| MALA | $106.668 \pm 29.601$ | $14.723 \pm 3.821$ | $4.061 \pm 1.587$ |
| AdaMALA | $211.948 \pm 133.363$ | $52.277 \pm 26.566$ | $6.401 \pm 3.344$ |
| HMC | $1624.773 \pm 544.777$ | $337.100 \pm 212.158$ | $6.052 \pm 2.062$ |
| mMALA | $6086.948 \pm 117.241$ | $5690.967 \pm 118.401$ | $5297.835 \pm 160.084$ |
| FisherMALA | $6437.419 \pm 207.548$ | $5981.960 \pm 156.072$ | $5628.541 \pm 168.425$ |
| *Caravan* ($d = 87$) | | | |
| MALA | $27.247 \pm 7.554$ | $5.890 \pm 0.398$ | $2.906 \pm 0.150$ |
| AdaMALA | $41.522 \pm 9.343$ | $7.144 \pm 0.663$ | $3.144 \pm 0.135$ |
| HMC | $787.901 \pm 173.863$ | $37.303 \pm 6.808$ | $4.238 \pm 0.532$ |
| mMALA | $179.899 \pm 61.502$ | $121.867 \pm 41.801$ | $51.414 \pm 24.995$ |
| FisherMALA | $2257.737 \pm 45.289$ | $1920.903 \pm 55.821$ | $\mathbf{498.016} \pm 96.692$ |
| *MNIST* ($d = 785$) | | | |
| MALA | $34.074 \pm 4.977$ | $7.589 \pm 0.149$ | $2.944 \pm 0.066$ |
| AdaMALA | $62.301 \pm 9.203$ | $8.188 \pm 0.399$ | $2.985 \pm 0.089$ |
| HMC | $889.386 \pm 118.050$ | $303.345 \pm 10.976$ | $114.439 \pm 20.965$ |
| mMALA | $51.589 \pm 3.447$ | $20.222 \pm 1.259$ | $5.240 \pm 0.492$ |
| FisherMALA | $1053.455 \pm 35.680$ | $811.522 \pm 19.165$ | $\mathbf{439.580} \pm 52.800$ |

Caravan dataset which visualizes chain autocorrelation. From all these results we can conclude that FisherMALA is better than all other samplers, and remarkably it outperforms significantly the position-dependent mMALA, especially in the high dimensional Caravan and MNIST datasets.

# 6  Related Work

There exist works that use some form of global preconditioning in gradient-based samplers for specialized targets such as latent Gaussian models [13, 44], which make use of the tractable Gaussian prior. Our method differs, since it is more agnostic to the target and learns a preconditioning from the history of gradients, analogously to how traditional adaptive MCMC learns from states [21, 38].

Several research works use position-dependent preconditioning $A(x)$ within gradient-based samplers, such as MALA. This is for example the idea behind Riemannian manifold MALA [20] and extensions [45]. Similar to Riemannian manifold methods there are approaches inspired by second order optimization that use the Hessian matrix, or some estimate of the Hessian, for sampling in a MALA-style manner [32, 17, 27]. Recently, such samplers and their time-continuous diffusion limits have been theoretically analyzed by obtaining convergence guarantees [12, 26]. All such methods form a position-dependent preconditioning and not the preconditioning we use in this paper, e.g. note that $\mathcal{I}^{-1}$ we consider here requires an expectation under the target and thus it is always a global preconditioner rather than a position-dependent one. Another difference is that our method has quadratic cost, while position-dependent preconditioning methods have cubic cost and they require computationally demanding quantities like the Hessian matrix. Therefore, in order for these methods to run faster some approximation may be needed, e.g. low rank [27] or quasi-Newton type [46, 24]. Furthermore, the Bayesian logistic results in Table 1 (see also Figure 3) show that the proposed FisherMALA method significantly outperforms manifold MALA [20] in Caravan and MNIST examples, despite the fact that manifold MALA preconditions with the exact negative inverse Hessian matrix of the log target. This could suggest that position-dependent preconditioning may be less effective in certain type of high-dimensional and log-concave problems.

Finally, there is recent work for learning flexible MCMC proposals by using neural networks [42, 25, 23, 41] and by adapting parameters using differentiable objectives [25, 29, 43, 14]. Our method differs, since it does not use objective functions (which have extra cost because they require an optimization to run in parallel with the MCMC chain), but instead it adapts similarly to traditionally MCMC methods by accumulating information from the observed history of the chain.

## 7   Conclusion

We derived an optimal preconditioning for the Langevin diffusion by optimizing the expected squared jumped distance, and subsequently we developed an adaptive MCMC algorithm that approximates the optimal preconditioning by applying an efficient quadratic cost recursion. Some possible topics for future research are: Firstly, it would be useful to investigate whether the score function differences that we use as the adaptation signal introduce any bias in the estimation of the inverse Fisher matrix. Secondly, it would be interesting to extend our method to learn the preconditioning for other gradient-based samplers such as Hamiltonian Monte Carlo (HMC), where such a matrix there is referred to as the inverse mass matrix. For HMC this is more complex since both the mass matrix and its inverse are needed in the iteration. Finally, it could be interesting to investigate adaptive schemes of the inverse Fisher matrix by using multiple parallel and interacting chains, similarly to ensemble covariance matrix estimation for Langevin diffusions [16].

## Acknowledgments and Disclosure of Funding

We are grateful to the reviewers for their comments. Also, we wish to thank Arnaud Doucet, Sam Power, Francisco Ruiz, Jiaxin Shi, Yee Whye Teh, Siran Liu, Kazuki Osawa and James Martens for useful discussions.

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
