# A Proofs

## A.1 Proof of Proposition 1

The difference between the logarithm of the backward and forward proposals of preconditioned MALA, i.e. the quantity $\log q(x_n|y_n) - \log q(y_n|x_n)$ can be written (ignoring the normalizing constants of the Gaussians which trivially cancel out) as,

$$-\frac{1}{2\sigma^2}\left(x_n - y_n - \frac{\sigma^2}{2}A\nabla\log\pi(y_n)\right)^\top A^{-1}\left(x_n - y_n - \frac{\sigma^2}{2}A\nabla\log\pi(y_n)\right)$$

$$+\frac{1}{2\sigma^2}\left(y_n - x_n - \frac{\sigma^2}{2}A\nabla\log\pi(x_n)\right)^\top A^{-1}\left(y_n - x_n - \frac{\sigma^2}{2}A\nabla\log\pi(x_n)\right). \quad (18)$$

Observe that the term $\frac{1}{2\sigma^2}(x_n - y_n)^\top A^{-1}(x_n - y_n)$ cancels out since it appears twice with opposite sign. The remaining terms after some simple algebra simplify as

$$\frac{1}{2}\left(x_n - y_n - \frac{\sigma^2}{4}A\nabla\log\pi(y_n)\right)^\top\nabla\log\pi(y_n) - \frac{1}{2}\left(y_n - x_n - \frac{\sigma^2}{4}A\nabla\log\pi(x_n)\right)^\top\nabla\log\pi(x_n)$$

$$= h(x_n, y_n) - h(y_n, x_n) \quad (19)$$

which completes the proof.

## A.2 Proof of Proposition 2

We assume $x_t \sim \pi(x_t)$. Then by taking the expectation of the r.h.s. of Eq. (4) (where the expectation is taken w.r.t. $x_t$ and the independent Brownian motion increment $B_{t+\delta} - B_t \sim \mathcal{N}(0, \delta I_d)$) and noting that $\mathbb{E}_{\pi(x_t)}[\nabla\log\pi(x_t)] = 0$ and $\mathbb{E}[B_{t+\delta} - B_t] = 0$ we conclude that $\mathbb{E}[x_{t+\delta} - x_t] = 0$. Then the covariance is

$$\mathbb{E}[(x_{t+\delta} - x_t)(x_{t+\delta} - x_t)^\top] =$$

$$= \mathbb{E}\left[(\frac{\delta}{2}A\nabla\log\pi(x_t) + \sqrt{A}(B_{t+\delta} - B_t))(\frac{\delta}{2}A\nabla\log\pi(x_t) + \sqrt{A}(B_{t+\delta} - B_t))^\top\right]$$

$$= \frac{\delta^2}{4}A\mathbb{E}_{\pi(x_t)}[\nabla\log\pi(x_t)\nabla\log\pi(x_t)^\top]A + \delta A$$

$$= \frac{\delta^2}{4}A\mathcal{I}A + \delta A,$$

where we used that $\mathbb{E}[(B_{t+\delta} - B_t)(B_{t+\delta} - B_t)^\top] = \delta I_d$, $\sqrt{A}\sqrt{A}^\top = A$ and that the cross covariance terms are zero.

## A.3 Proof of Proposition 3

The expected squared jumped distance is written as

$$J(\delta, A) = \text{tr}\left(\frac{\delta^2}{4}A\mathcal{I}A + \delta A\right) = \frac{\delta^2}{4}\text{tr}(A\mathcal{I}A) + \delta c,$$

where we used the constraint $\text{tr}(A) = c$. Since $c$ is just a constant to minimize $J(\delta, A)$ is the same as minimizing $\text{tr}(A\mathcal{I}A)$, a quadratic convex loss since $\mathcal{I}$ is positive definite, under the constraint that $A$ is symmetric positive definite matrix and $\text{tr}(A) = c$. To deal with the equality constraint we consider the Lagrangian

$$\text{tr}(A\mathcal{I}A) - \lambda(\text{tr}(A) - c).$$

By taking derivatives wrt the matrix $A$ (using the matrix derivative identities $\frac{\partial}{\partial X}\text{tr}(XBX) = X^\top B^\top + B^\top X^\top$ and $\frac{\partial}{\partial X}\text{tr}(X) = I_d$ for arbitrary $d \times d$ square matrices $X, B$) and setting to zero we see that $A$ must satisfy the linear equation

$$A^\top\mathcal{I} + \mathcal{I}A^\top = \lambda I_d,$$

where we used that $\mathcal{I}$ is a symmetric matrix. This is a set of linear equations and given that each eigenvalue $\mu_i$ of $\mathcal{I}$ satisfies $0 < \mu_i < \infty$, so that $\mathcal{I}$ is invertible, there is an unique solution given by $A = (1/2)\lambda\mathcal{I}^{-1}$. The Lagrange multiplier $\lambda$ is chosen so that $\text{tr}(A) = c$ which leads to the optimal $A^*$

$$A^* = \frac{c}{\sum_{i=1}^{d}\frac{1}{\mu_i}}\mathcal{I}^{-1}.$$

Note that $A^*$ turned out to be symmetric and positive definite as desired. For this $A^*$ the optimal loss value is $\text{tr}(A^*\mathcal{I}A^*) = \frac{c^2}{\sum_{i=1}^{d}\frac{1}{\mu_i}}$, for which we further need to disambiguate whether this is the global minimum or maximum. We can do this by choosing a different matrix that satisfies the constraint $\text{tr}(A) = c$ and compare its loss with the optimal loss $\frac{c^2}{\sum_{i=1}^{d}\frac{1}{\mu_i}}$. For example, one such matrix is $A = \frac{c}{d}I_d$, which has loss value $\frac{c^2(\sum_{i=1}^{d}\mu_i)}{d^2}$. Then by using the Cauchy-Schwarz inequality $d^2 = (\sum_{i=1}^{d}\frac{\sqrt{\mu_i}}{\sqrt{\mu_i}})^2 \leq (\sum_{i=1}^{d}\mu_i)(\sum_{i=1}^{d}\frac{1}{\mu_i})$ we obtain $\frac{c^2(\sum_{i=1}^{d}\mu_i)}{d^2} \geq \frac{c^2(\sum_{i=1}^{d}\mu_i)}{(\sum_{i=1}^{d}\mu_i)(\sum_{i=1}^{d}\frac{1}{\mu_i})} = \frac{c^2}{\sum_{i=1}^{d}\frac{1}{\mu_i}}$. This shows that $A^*$ achieves the global minimum which completes the proof.

### A.4 Proof of Proposition 4

We first state and prove the following intermediate result.

**Lemma 1.** *Suppose the positive definite matrix $I_d - zz^\top$ where $z \in \mathbb{R}^d$ and $z^\top z \leq 1$. Then, a square root matrix $R$, satisfying $RR^\top = A$, has the form $R = I_d - rzz^\top$ where $r = \frac{1}{1+\sqrt{1-z^\top z}}$.*

*Proof.* We hypothesize that $R$ has the form $I_d - rzz^\top$ for some scalar $r$. Then since $RR^\top = I_d - zz^\top$ we see that $r$ must satisfy the quadratic equation $r^2 z^\top z - 2r + 1 = 0$, which has two real solutions $\frac{1\pm\sqrt{1-z^\top z}}{z^\top z}$ and we will use $\frac{1-\sqrt{1-z^\top z}}{z^\top z} \leq 1$ which ensures $R$ is positive definite. This solution can also be written as $r = \frac{1}{1+\sqrt{1-z^\top z}}$. $\square$

To prove the proposition we need to find a square root matrix $R_1$ of $A_1 = \frac{1}{\lambda}\left(I_d - \frac{s_1 s_1^\top}{\lambda+s_1^\top s_1}\right)$ where we clearly need to specify a square root matrix for $I_d - \frac{s_1 s_1^\top}{\lambda+s_1^\top s_1}$. We observe that by setting $z = \frac{s_1}{\sqrt{\lambda+s_1^\top s_1}}$ Lemma 1 is applicable so that the square root matrix is

$$R_1 = \frac{1}{\sqrt{\lambda}}\left(I_d - r_1\frac{s_1 s_1^\top}{\lambda+s_1^\top s_1}\right), \quad r_1 = \frac{1}{1+\sqrt{\frac{\lambda}{\lambda+s_1^\top s_1}}}.$$

Similarly by applying again Lemma 1 we can find $R_n$ for any $n > 1$.

The computation of $R_n$ costs $O(d^2)$ per iteration. Firstly, the vector $\phi_n = R_{n-1}^\top s_n$ is computed which is a matrix-vector multiplication. The next step is to compute the scalar $r_n$ in $O(d)$ (involving the dot product $\phi_n^\top\phi_n$) and then the scaled vector $\phi_n' = \frac{r_n}{1+\phi_n^\top\phi_n}\phi_n$ also an $O(d)$ operation. Then we need two additional $O(d^2)$ multiplication operations to obtain firstly the vector $t_n = R_{n-1}\phi_n$ and secondly the outer vector product $t_n(\phi_n')^\top$. Finally, the update is $R_n = R_{n-1} - t_n(\phi_n')^\top$ which requires a final $O(d^2)$ addition operation of two matrices which is typically cheaper than $O(d^2)$ multiplication. Therefore, overall the cost is $O(d^2)$.

## B Generalizing the recursion over arbitrary learning rate sequences

Suppose we have a sequence of learning rates $\gamma_1, \gamma_2, \ldots,$. Then a stochastic approximation of the Fisher matrix $\mathcal{I}$ takes the form

$$\mathcal{I}_n = \mathcal{I}_{n-1} + \gamma_n(s_n s_n^\top - \mathcal{I}_{n-1}) = (1-\gamma_n)\mathcal{I}_{n-1} + \gamma_n s_n s_n^\top,$$

where the sequence is initialized at $\mathcal{I}_1 = s_1 s_1^\top + \lambda I$. The inverse of the empirical Fisher is written as

$$A_n = \left((1-\gamma_n)\mathcal{I}_{n-1} + \gamma_n s_n s_n^\top\right)^{-1} = \frac{1}{1-\gamma_n}\left(A_{n-1} - \frac{A_{n-1}s_n s_n^\top A_{n-1}}{\frac{1-\gamma_n}{\gamma_n} + s_n^\top \mathcal{I}_{n-1}^{-1} s_n}\right),$$

which is initialized at $A_1 = \frac{1}{\lambda}\left(I_d - \frac{s_1 s_1^\top}{\lambda + s_1^\top s_1}\right)$ for which the square root $R_1$ is the same as for the standard learning rate $\gamma_n = 1/n$. The square root recursion for $n > 1$ takes the form

$$R_n = \frac{1}{\sqrt{1-\gamma_n}}\left(R_{n-1} - r_n \frac{(R_{n-1}\phi_n)\phi_n^\top}{(1-\gamma_n)/\gamma_n + \phi_n^\top \phi_n}\right), \ \phi_n = R_n^\top s_n, \ r_n = \frac{1}{1 + \sqrt{\frac{(1-\gamma_n)/\gamma_n}{(1-\gamma_n)/\gamma_n + \phi_n^\top \phi_n}}}.$$

## C   FisherMALA with paired mean and covariance stochastic approximation

Here, we derive a recursion for the empirical Fisher that centers the score function vectors using the standard procedure by recursively estimating also the mean. We start from the following consistent estimator of the inverse Fisher:

$$A_n = \left(\frac{1}{n-1}\sum_{i=1}^n (s_i - \bar{s}_n)(s_i - \bar{s}_n)^\top + \frac{\lambda}{n-1}I_d\right)^{-1},$$

where $\bar{s}_n = \frac{1}{n}\sum_{i=1}^n s_i$. This follows the recursion

$$A_n = \left(\frac{n-2}{n-1}A_{n-1}^{-1} + \frac{1}{n}\delta_n\delta_n^\top\right)^{-1} = \frac{n-1}{n-2}A_{n-1} - \frac{(n-1)^2}{(n-2)^2}\frac{A_{n-1}\delta_n\delta_n^\top A_{n-1}}{n + \frac{n-1}{n-2}\delta_n^\top A_{n-1}\delta_n}$$

$$= \frac{1}{\lambda_{n-1}}\left(A_{n-1} - \frac{A_{n-1}\delta_n\delta_n^\top A_{n-1}}{n\lambda_{n-1} + \delta_n^\top A_{n-1}\delta_n}\right).$$

Here, $\delta_n = s_n - \bar{s}_{n-1}$ and we defined the sequence of scalars $\lambda_n = \frac{n-1}{n}$, for $n \geq 2$ while the starting point of this sequence $n = 1$ we define it to be equal to the parameter parameter $\lambda$, i.e. $\lambda_1 = \lambda > 0$. The recursion starts at $A_2$ given by

$$A_2 = \left(\frac{1}{2}\delta_2\delta_2^\top + \lambda_1 I\right)^{-1} = \frac{1}{\lambda_1}\left(I_d - \frac{\delta_2\delta_2^\top}{2\lambda_1 + \delta_2^\top \delta_2}\right),$$

where $\delta_2 = s_2 - s_1$. Along with the above we recursively estimate also the mean vector (for $n \geq 1$): $\bar{s}_n = \frac{n-1}{n}\bar{s}_{n-1} + \frac{1}{n}s_n$.

To express a recursion of square root matrix, such that $A_n = R_n R_n^\top$ we first write

$$A_n = \frac{1}{\lambda_{n-1}}R_{n-1}\left(I_d - \frac{R_{n-1}^\top \delta_n\delta_n^\top R_{n-1}}{n\lambda_{n-1} + \delta_n^\top A_{n-1}\delta_n}\right)R_{n-1}^\top$$

$$= \frac{1}{\lambda_{n-1}}R_{n-1}\left(I_d - \frac{\phi_n\phi_n^\top}{n\lambda_{n-1} + \phi_n^\top \phi_n}\right)R_{n-1}^\top.$$

Then we can recognize the square root recursion as

$$R_n = \frac{1}{\sqrt{\lambda_{n-1}}}R_{n-1}\left(I_d - r_n\frac{\phi_n\phi_n^\top}{n\lambda_{n-1} + \phi_n^\top \phi_n}\right), \ \ r_n = \frac{1}{1 + \sqrt{\frac{n\lambda_{n-1}}{n\lambda_{n-1} + \phi_n^\top \phi_n}}},$$

which is initialized at $R_2 = \frac{1}{\sqrt{\lambda_1}}\left(I_d - r_2\frac{\delta_2\delta_2^\top}{2\lambda_1 + \delta_2^\top \delta_2}\right), \ r_2 = \frac{1}{1 + \sqrt{\frac{2\lambda_1}{2\lambda_1 + \delta_2^\top \delta_2}}}.$

## D   Initialization of AdaMALA

To initialize AdaMALA we first perform $n_0 = 500$ iterations with simple MALA where we adapt the step size parameter $\sigma^2$. Thus, this part of the initialization is exactly the same used by FisherMALA. However, for AdaMALA we do an additional set of $n_0 = 500$ iterations where simple MALA still runs and collects samples which are used to sequentially update the empirical covariance matrix $\Sigma_n$. The purpose of this second phase is to play the role of "warm-up" and provide a reasonable initialization for $\Sigma_n$. After the second phase (so in total 1000 iterations) AdaMALA starts running having as a preconditioner $\Sigma_n$, which keeps adapted in every iteration until the last burn-in iteration.

# E  Additional results

## E.1  The step size $\sigma^2$ is maximized when preconditioning becomes effective

To experimentally backup our claims in Section 3 that the discretization step size, denoted there by $\delta$ or $\sigma^2$, gets large when the preconditioner is selected efficiently, in Figure 4 we report the final learned values (after burn-in adaptation iterations) of $\sigma^2$ for MALA, AdaMALA and FisherMALA. For all these three algorithms the values of $\sigma^2$ are comparable because all use an overall preconditioning of the form $\frac{\sigma^2}{\frac{1}{d}\operatorname{tr}(A)} A$ and only the matrix $A$ is changing among them. For example, simple MALA sets this matrix to $A = I_d$, while AdaMALA and FisherMALA use their own procedures to learn more complex matrices. Figure 4 shows the estimated $\sigma^2$, for the four datasets reported in the main text in Table 1. This shows that FisherMALA achieves significantly larger $\sigma^2$ in all cases, which can be orders of magnitude larger than the two other algorithms (note the $y$ axis in Figure 4 is in log scale).

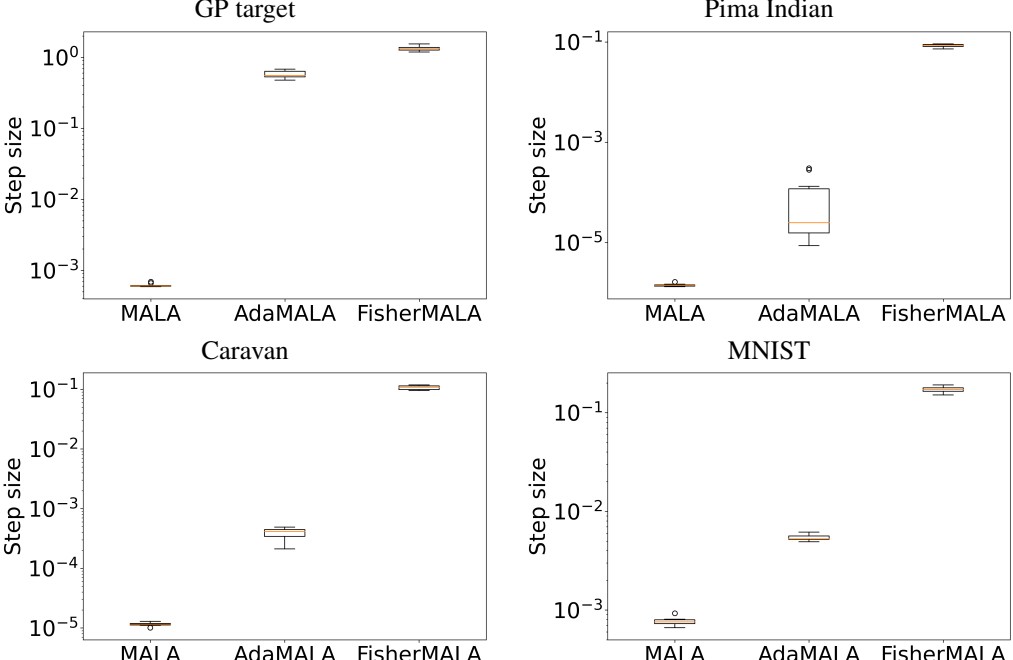

Figure 4: It shows the estimated values of $\sigma^2$ for MALA, AdaMALA and FisherMALA using boxplots (each computed from the 10 random repeats; see Table 1) for the four datasets presented in Table 1. For better visibility the $y$ axis is shown in log scale.

## E.2  Additional plots and tables

Figure 5 and 6 display additional visualizations for the 2-D Gaussian and the GP target experiments. Tables 2-6 provide the ESS scores for the inhomogeneous Gaussian target and all remaining Bayesian logistic regression datasets, that were not included in the main paper. Bold font in the "Min ESS" entry in the tables indicates statistical significance. Similarly, Figures 7-14 show the log target values across iterations for the four best samplers, i.e. excluding simple MALA which is the least performing method.

## E.3  The effect of Raoblackwellization and comparison with paired stochastic estimation

Finally, we compare three versions of FisherMALA: (i) The one that uses the Raoblackwellized signal $s_n^\delta$ from Eq. (16), which is our main proposed method used in the main paper and all previous results (in this section we will denote this as FisherMALA-with-RB), (ii) the one that uses the initial score function difference from Eq. (15) (FisherMALA-no-RB) and (iii) and FisherMALA with paired mean and covariance stochastic estimation (FisherMALA-paired-est) as descibed in Appendix C. Table

Table 2: ESS scores for the inhomogeneous Gaussian target.

|  | Max ESS | Median ESS | Min ESS |
| --- | --- | --- | --- |
| MALA | $13695.291 \pm 1369.515$ | $9.793 \pm 0.655$ | $2.943 \pm 0.130$ |
| AdaMALA | $4310.690 \pm 606.618$ | $70.802 \pm 14.912$ | $9.225 \pm 3.272$ |
| HMC | $19362.103 \pm 1372.400$ | $381.205 \pm 101.781$ | $42.033 \pm 33.080$ |
| mMALA | $2354.354 \pm 65.835$ | $2014.801 \pm 23.713$ | $1490.119 \pm 108.745$ |
| FisherMALA | $2347.340 \pm 70.234$ | $2002.579 \pm 30.001$ | $1500.983 \pm 67.087$ |

Table 3: ESS scores for the Heart dataset.

|  | Max ESS | Median ESS | Min ESS |
| --- | --- | --- | --- |
| MALA | $68.774 \pm 25.304$ | $5.354 \pm 1.056$ | $2.898 \pm 0.104$ |
| AdaMALA | $208.636 \pm 124.762$ | $14.762 \pm 9.134$ | $3.781 \pm 0.731$ |
| HMC | $387.321 \pm 311.673$ | $12.991 \pm 4.009$ | $4.064 \pm 1.120$ |
| mMALA | $878.858 \pm 1079.674$ | $789.356 \pm 969.806$ | $651.793 \pm 806.477$ |
| FisherMALA | $4864.278 \pm 103.277$ | $4474.288 \pm 102.029$ | $\mathbf{3954.793} \pm 199.832$ |

Table 4: ESS scores for the German Credit dataset.

|  | Max ESS | Median ESS | Min ESS |
| --- | --- | --- | --- |
| MALA | $262.206 \pm 211.839$ | $5.932 \pm 0.668$ | $2.972 \pm 0.212$ |
| AdaMALA | $223.592 \pm 111.914$ | $16.111 \pm 5.058$ | $3.774 \pm 0.653$ |
| HMC | $10439.824 \pm 9572.157$ | $45.872 \pm 7.823$ | $5.431 \pm 1.257$ |
| mMALA | $3066.605 \pm 100.768$ | $2767.022 \pm 94.222$ | $2342.902 \pm 112.610$ |
| FisherMALA | $3951.807 \pm 78.858$ | $3582.184 \pm 90.551$ | $\mathbf{3011.483} \pm 258.154$ |

Table 5: ESS scores for the Australian Credit dataset.

|  | Max ESS | Median ESS | Min ESS |
| --- | --- | --- | --- |
| MALA | $15.627 \pm 12.892$ | $3.823 \pm 1.166$ | $2.611 \pm 0.538$ |
| AdaMALA | $1525.373 \pm 1600.986$ | $6.986 \pm 3.200$ | $3.297 \pm 0.456$ |
| HMC | $1282.235 \pm 932.038$ | $6.966 \pm 1.249$ | $2.856 \pm 0.095$ |
| mMALA | $2609.462 \pm 881.967$ | $2308.175 \pm 776.872$ | $1869.364 \pm 630.880$ |
| FisherMALA | $4732.724 \pm 116.074$ | $4361.969 \pm 104.750$ | $\mathbf{3772.086} \pm 265.170$ |

Table 6: ESS scores for the Ripley dataset.

|  | Max ESS | Median ESS | Min ESS |
| --- | --- | --- | --- |
| MALA | $2058.325 \pm 180.839$ | $496.981 \pm 68.029$ | $427.492 \pm 60.006$ |
| AdaMALA | $9678.793 \pm 384.295$ | $9497.814 \pm 463.059$ | $9272.026 \pm 412.361$ |
| HMC | $18403.796 \pm 3202.136$ | $18254.161 \pm 3513.550$ | $7644.709 \pm 2288.559$ |
| mMALA | $9333.633 \pm 280.238$ | $8941.579 \pm 288.223$ | $8655.640 \pm 396.106$ |
| FisherMALA | $9875.968 \pm 218.801$ | $9673.009 \pm 280.759$ | $9244.631 \pm 559.137$ |

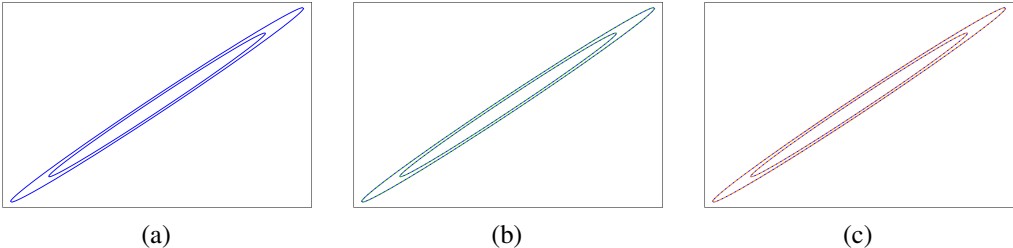

(a)                                    (b)                                    (c)

Figure 5: Panel (a) shows the true covariance of the 2-D Gaussian. Panel (b) shows the estimated covariance by FisherMALA (dashed green line), where for comparison the true covariance is also shown in blue. Panel (c) shows the estimated covariance by AdaMALA (dashed red line).

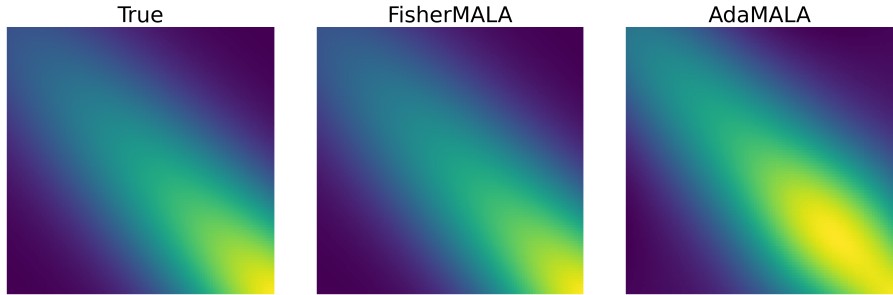

Figure 6: The covariance matrices for the GP target, where in the right panel is the covariance estimated by AdaMALA which was not displayed in Figure 1 in the main text.

7 compares the three versions of FisherMALA in terms of ESS for all problems, which shows that FisherMALA-paired-est is significantly worse than the other two methods that learn based on score function increments. These two latter methods, FisherMALA-with-RB and FisherMALA-no-RB, have similar performance without significant difference (the highest difference in terms of Min ESS is in Pima Indians dataset, but still not statistically significant).

Figure 15 displays the Frobenius norms for FisherMALA with Raoblackwellization and FisherMALA without Raoblackwellization in the two 100-dimensional Gaussian targets. It shows that the Raoblackwellized signal $s_n^\delta$ leads to slightly faster convergence, which agrees with the theory that says that Raoblackwellization should reduce the variance.

Finally, Table 8 reports numerical performance of the non-centered version of FisherMALA where we learn directly from the score function vectors $s_n$, i.e. without centering or using score function increments. From this table we can see that FisherMALA (non-centered) performs worse than the other FisherMALA variants, and only on Ripley dataset works equally well with the rest.

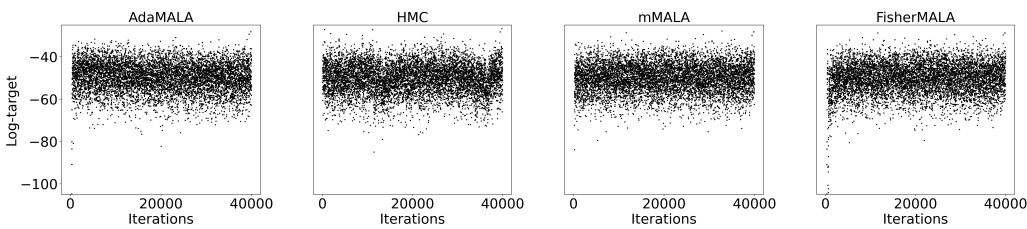

Figure 7: The evolution of the log-target across iterations in the GP target.

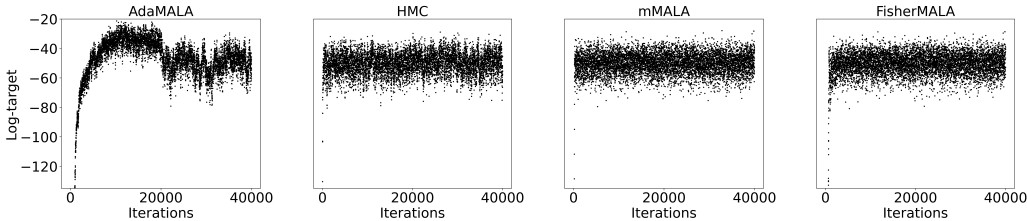

Figure 8: The evolution of the log-target across iterations in the inhomogeneous Gaussian target.

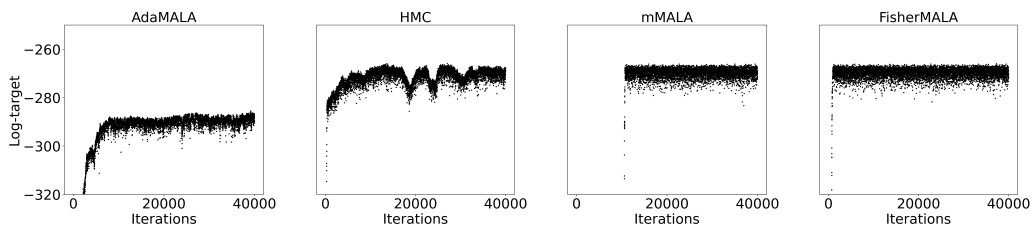

Figure 9: The evolution of the log-target across iterations in Pima Indians dataset.

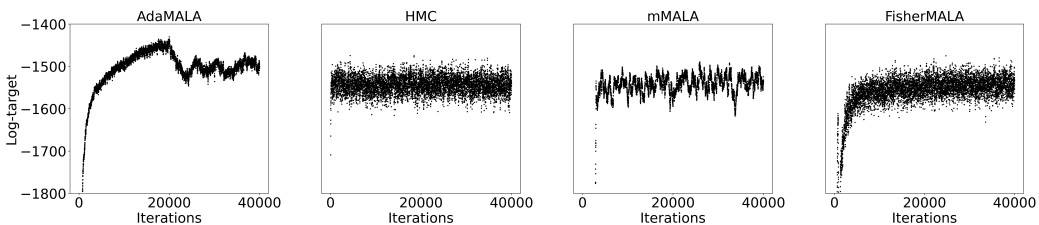

Figure 10: The evolution of the log-target across iterations in MNIST dataset.

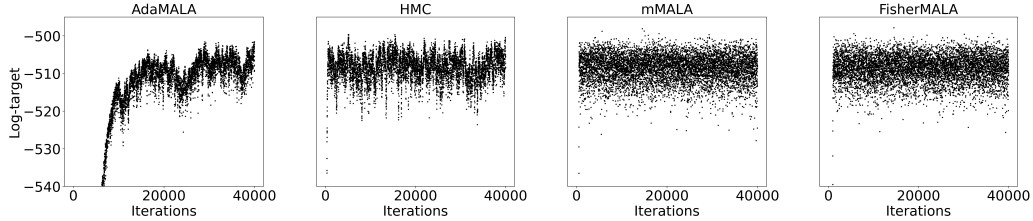

Figure 11: The evolution of the log-target across iterations in German Credit dataset.

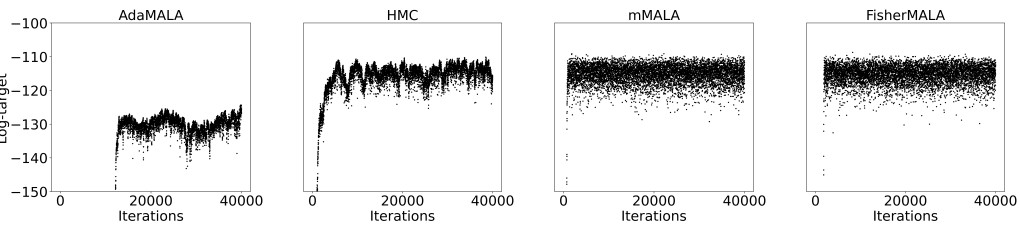

Figure 12: The evolution of the log-target across iterations in Heart dataset.

Table 7: Comparison of ESS scores for three versions of FisherMALA: the first with Raoblackwellized score function differences in (16), the second based on the initial adaptation signal of score function differences from (15), and the third based on paired stochastic estimation.

| | Max ESS | Median ESS | Min ESS |
|---|---|---|---|
| *GP target* | | | |
| FisherMALA-with-RB | $2096.259 \pm 94.751$ | $1923.753 \pm 95.820$ | $1784.962 \pm 104.440$ |
| FisherMALA-no-RB | $2064.940 \pm 87.943$ | $1916.990 \pm 85.208$ | $1794.114 \pm 103.711$ |
| FisherMALA-paired-est | $1802.141 \pm 142.784$ | $1583.570 \pm 109.241$ | $1226.303 \pm 244.752$ |
| *Inhomog. Gaussian* | | | |
| FisherMALA-with-RB | $2347.340 \pm 70.234$ | $2002.579 \pm 30.001$ | $1500.983 \pm 67.087$ |
| FisherMALA-no-RB | $2351.481 \pm 78.894$ | $2012.243 \pm 30.024$ | $1489.617 \pm 133.619$ |
| FisherMALA-paired-est | $1941.994 \pm 106.710$ | $1147.138 \pm 61.591$ | $109.160 \pm 57.998$ |
| *Heart* | | | |
| FisherMALA-with-RB | $4864.278 \pm 103.277$ | $4474.288 \pm 102.029$ | $3954.793 \pm 199.832$ |
| FisherMALA-no-RB | $4893.063 \pm 107.068$ | $4455.591 \pm 98.542$ | $3977.741 \pm 194.922$ |
| FisherMALA-paired-est | $4804.365 \pm 176.747$ | $2519.187 \pm 693.945$ | $441.434 \pm 386.287$ |
| *German Credit* | | | |
| FisherMALA-with-RB | $3951.807 \pm 78.858$ | $3582.184 \pm 90.551$ | $3011.483 \pm 258.154$ |
| FisherMALA-no-RB | $3979.744 \pm 79.647$ | $3616.894 \pm 104.722$ | $3031.384 \pm 228.345$ |
| FisherMALA-paired-est | $3960.773 \pm 105.169$ | $3097.557 \pm 252.619$ | $397.034 \pm 244.768$ |
| *Australian Credit* | | | |
| FisherMALA-with-RB | $4732.724 \pm 116.074$ | $4361.969 \pm 104.750$ | $3772.086 \pm 265.170$ |
| FisherMALA-no-RB | $4711.549 \pm 115.329$ | $4364.347 \pm 95.004$ | $3790.949 \pm 253.464$ |
| FisherMALA-paired-est | $4887.606 \pm 173.626$ | $3603.765 \pm 725.018$ | $84.202 \pm 44.750$ |
| *Ripley* | | | |
| FisherMALA-with-RB | $9875.968 \pm 218.801$ | $9673.009 \pm 280.759$ | $9244.631 \pm 559.137$ |
| FisherMALA-no-RB | $9852.895 \pm 281.295$ | $9679.384 \pm 303.946$ | $9272.040 \pm 581.732$ |
| FisherMALA-paired-est | $9869.053 \pm 321.031$ | $9598.430 \pm 330.766$ | $9217.330 \pm 584.224$ |
| *Pima Indians* | | | |
| FisherMALA-with-RB | $6437.419 \pm 207.548$ | $5981.960 \pm 156.072$ | $5628.541 \pm 168.425$ |
| FisherMALA-no-RB | $6448.999 \pm 199.817$ | $5977.292 \pm 122.852$ | $5585.217 \pm 160.586$ |
| FisherMALA-paired-est | $6048.419 \pm 650.262$ | $2618.271 \pm 889.425$ | $788.687 \pm 388.978$ |
| *Caravan* | | | |
| FisherMALA-with-RB | $2257.737 \pm 45.289$ | $1920.903 \pm 55.821$ | $498.016 \pm 96.692$ |
| FisherMALA-no-RB | $2241.262 \pm 47.873$ | $1908.045 \pm 62.430$ | $509.913 \pm 115.563$ |
| FisherMALA-paired-est | $1930.109 \pm 208.848$ | $1107.987 \pm 83.439$ | $87.456 \pm 90.858$ |
| *MNIST* | | | |
| FisherMALA-with-RB | $1053.455 \pm 35.680$ | $811.522 \pm 19.165$ | $439.580 \pm 52.800$ |
| FisherMALA-no-RB | $1036.138 \pm 32.399$ | $803.210 \pm 16.163$ | $437.325 \pm 40.040$ |
| FisherMALA-paired est | $301.055 \pm 37.597$ | $13.819 \pm 1.127$ | $3.176 \pm 0.113$ |

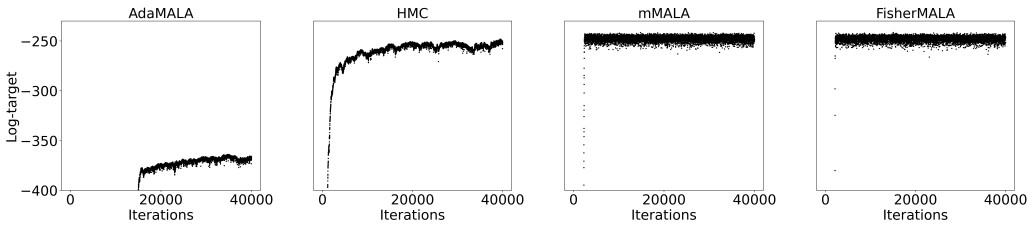

Figure 13: The evolution of the log-target across iterations in Australian Credit dataset.

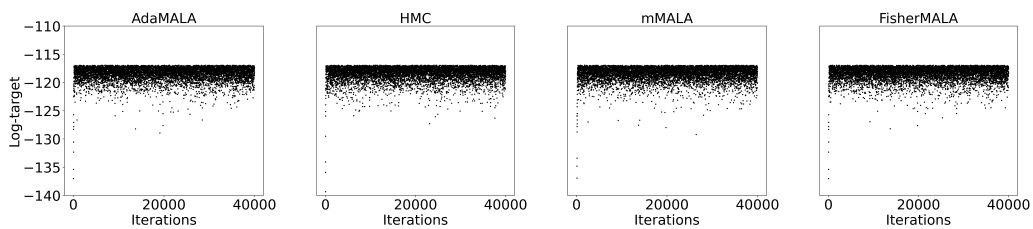

Figure 14: The evolution of the log-target across iterations in Ripley dataset.

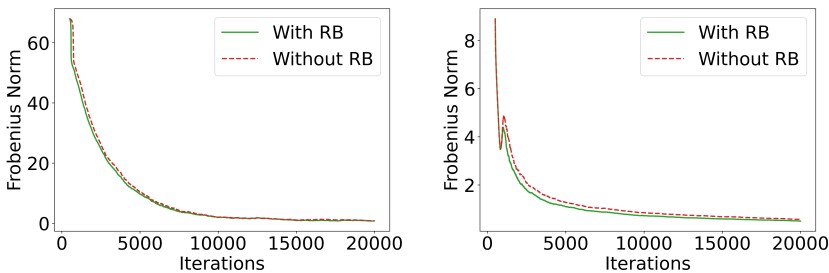

Figure 15: The effect of Raoblackwellization. Left panel shows the evolution of the Frobenius norm in the GP target and right panel for the inhomogeneous Gaussian target.

Table 8: Performance of FisherMALA (non-centered), in a subset of the targets, which learns directly from the score function vectors $s_n$.

|  | Max ESS | Median ESS | Min ESS |
|---|---|---|---|
| *GP target*
FisherMALA (non-centered) | $1740.943 \pm 157.871$ | $518.924 \pm 579.639$ | $48.218 \pm 117.349$ |
| *Ripley*
FisherMALA (non-centered) | $9881.540 \pm 353.377$ | $9636.357 \pm 313.009$ | $9237.885 \pm 710.741$ |
| *Pima Indians*
FisherMALA (non-centered) | $5520.181 \pm 1781.518$ | $474.990 \pm 587.788$ | $65.313 \pm 59.316$ |
| *Caravan*
FisherMALA (non-centered) | $1602.723 \pm 164.497$ | $14.226 \pm 4.429$ | $3.298 \pm 0.141$ |
| *MNIST*
FisherMALA (non-centered) | $271.629 \pm 22.918$ | $22.147 \pm 1.683$ | $3.744 \pm 0.139$ |