# OpenReview forum: "Optimal Preconditioning and Fisher Adaptive Langevin Sampling"
_NeurIPS.cc/2023/Conference — NeurIPS 2023 poster_

### Official Review · Reviewer_Zcpk · 2023-06-14

**Soundness:** 2 fair
**Presentation:** 3 good
**Contribution:** 2 fair
**Rating:** 6
**Confidence:** 5

**Summary:**

The paper presents an “optimal” preconditioning for the Langevin diffusion by analytically maximizing the expected squared jumped distance. The authors have identified the optimal preconditioning as an inverse Fisher information covariance matrix. They apply this result to the Metropolis adjusted Langevin algorithm (MALA) and derive an efficient adaptive MCMC scheme that learns the preconditioning from the gradient history produced during the algorithm's execution. The authors show through experiments that their proposed algorithm can outperform other standard methods.

**Strengths:**

The paper introduces an innovative approach with the use of the inverse Fisher information covariance matrix as a preconditioner for MALA; to my knowledge this has not been previously proposed. The preconditioner is shown to maximize the expected squared jump distance, which could potentially improve the convergence of the algorithm.

The adaptive learning of the preconditioning from the history of gradients generated during the algorithm execution is a useful idea.

The experiments presented demonstrate the algorithm’s promise.


**Weaknesses:**

The paper lacks theoretical justification to support the empirical findings, which might raise questions about the universality and optimal applicability of the proposed method.

**Questions:**

The Fisher information preconditioner, which is the expectation of the negative Hessian of the log-density, is very reminiscent of the Newton–Langevin diffusion which uses the same but without the expectation; see S. Chewi, T. Le Gouic, C. Lu, T. Maunu, P. Rigollet, A. J. Stromme, "Exponential ergodicity of mirror-Langevin diffusion," and Appendix A.4 of R. Li, M. Tao, S. S. Vempala, A. Wibisono's work, "The mirror Langevin algorithm converges with vanishing bias." Please provide a comparison with this method, both in theory and practice.

The Newton–Langevin diffusion in particular enjoys convergence guarantees in continuous time that attest to its affine-invariant nature. Can anything be proven regarding convergence rates with this choice of preconditioner?

I did not see any claim that $\pi$ is the stationary distribution of Algorithm 1. Can the authors provide justification or proof for this?

**Limitations:**

Yes.

---

> ### Author Rebuttal · Authors · 2023-08-09
>
> Thank you for bringing the interesting mirror Langevin diffusion work to our attention, and we will discuss it in the related work. By reading the papers of S. Chewi et al and Li et al, we can observe that for Gaussian target $\pi(x)=\mathcal{N}(x|\mu, \Sigma)$ the discrete version of Newton Langevin Diffusion (NLD) is precisely the Riemannian manifold MALA (mMALA). This is because in such case  the NLD proposal (e.g. by discretizing the Equation 33 in Appendix A.4 in Li et al) reduces to
> $$
> x_{n+1} = x_n + \frac{\sigma^2}{2} \Sigma \nabla \log \pi(x_n) +
> \sigma \sqrt{\Sigma} \epsilon_n
> $$
> which is the mMALA from Girolami and Galderhead (2011). We will clarify that mMALA in such case reduces to NLD. For the logistic regression case mMALA uses as a preconditioner the negative Hessian matrix (see also Section 7 in Girolami and Calderhead (2011)).
> $$
> A(\theta)
> = Z^\top \Lambda(\theta) Z + I_d
> $$
> where $\Lambda(\theta)$ is a diagonal matrix with entries $\lambda_i = \sigma(\theta^\top z_i) (1 - \sigma(\theta^\top z_i))$. This is precisely the negative Hessian matrix of the log joint $- \frac{1}{2} || \theta ||^2 + \sum_{i=1}^m y_i \log  \sigma(\theta^\top z_i) (1 - \sigma(\theta^\top z_i))$ (see section 5.3 in our paper). We will clarify this by mentioning the connection with NLD. Therefore, for Bayesian logistic regression the mMALA that we have included in the comparison is very similar to NLD. Of course, NLD for this non-Gaussian example includes also an additional non-zero drift term that requires 3-order derivatives. While the numerical Newton iterative method, described in Appendix E.2 in S. Chewi et al, avoids 3-order derivatives it will still increase considerably the running time. For instance,
> note that mMALA alone which requires in every iteration a Choleksy decomposition of the Hessian matrix (an operation of cubic cost) is already more than twice slower than FisherMALA and the other MALA schemes in the largest MNIST dataset.
>
> Regarding the theoretical justification and proving convergence rates with this choice of preconditioner, reviewer's comment  is valid since our work does not provide such results. We would like to point out that our current work focuses on the methodological and computational/algorithmic developments. Theoretical analysis requires substantial more work which is beyond the scope of the current paper, and we plan to address it in the future.

---

> > ### Comment · Reviewer_Zcpk · 2023-08-11
> > **Response**
> >
> > Thank you for the response. I am satisfied with the rebuttal and I would like to raise my score to a 6.

---

### Official Review · Reviewer_K8aZ · 2023-06-28

**Soundness:** 3 good
**Presentation:** 4 excellent
**Contribution:** 3 good
**Rating:** 7
**Confidence:** 5

**Summary:**

The authors study a preconditioning matrix for the Langevin diffusion. It is given by maximizing the expected squared jumped distance. It turns out that the preconditioning is an inverse Fisher information covariance matrix for the target distribution. The authors apply the MALA scheme with this preconditioner to compute the MCMC algorithms. It is shown that in the Gaussian target distribution, this preconditioner performs as the Newton-type method. Numerical examples demonstrate the effectiveness of this method. The authors compared it with different methods, such as the position-dependent Riemannian manifold MALA sampler.


**Strengths:**

1. The authors present an important example of preconditioners, namely Fisher information matrix for Langevin dynamics.
It is motivated by the Gaussian target distribution or the expected squared jumped distance. The method is similar to Newton's method in the Gaussian case.

2. The numerical examples demonstrate the method's effectiveness for the Gaussian target distributions.



**Weaknesses:**

1. The computation of the inverse of Fisher-information may not be simple. The authors propose some modification methods to approximate it. Some computational complexity analysis is needed.

2. There is no theory supporting the convergence speed for non-Gaussian settings. Suppose the target distribution is a mixed Gaussian or a distribution that is non-log concave. Can the authors provide some numerical examples for these non-Gaussian target distributions to test the algorithm's performance?  It would be curious to know how Fisher information preconditioners behave.

3.  There is some related literature in this direction. One picks different choices of preconditioners motivated by the Gaussian target distributions.

Alfredo Garbuno-Inigo, et al. Interacting Langevin Diffusions: Gradient Structure And Ensemble Kalman Sampler, SIAM Journal on applied dynamical sysetm, 2019.
Yifei Wang, et.al. Projected Wasserstein gradient descent for high-dimensional Bayesian inference, SIAM/ASA Journal on Uncertainty Quantification, 2022.
Yifei Wang, et.al. Accelerated Information Gradient Flow, Journal of Scientific computing, 2021


**Questions:**

The authors also discuss the Hamiltonian Monte-Carlo methods with Fisher information preconditioners. Can authors briefly discuss the motivation why the Hamiltonian Monte-Carlo with preconditioners will behave better than the overdamped Langevin dynamics? Would you explain any potential advantages for the Hamiltonian Monte-Carlo methods, in terms of convergence analysis?

**Limitations:**

There are no limitations.

---

> ### Author Rebuttal · Authors · 2023-08-09
>
> Regarding the iterative computation of the inverse Fisher matrix, we will add the complexity analysis in Appendix A.4 (where we prove Proposition 4) to fully explain the $O(d^2)$ cost per iteration.  To describe here the main part of this analysis note that the main adaptive computation is the update of the square root of the Fisher inverse in Equation (13). This is done in the following order. Firstly,  the vector $\phi_n = R_{n-1}^\top s_n$ is computed which is a matrix-vector multiplication with $O(d^2)$ cost. The next step is to compute the scalar $r_n$ in $O(d)$ (involving the dot product $\phi_n^\top \phi_n$) and then the scaled vector $\phi_n' = \frac{r_n}{1 + \phi_n^\top \phi_n} \phi_n$ also an $O(d)$ operation. Then we need two additional $O(d^2)$ multiplication operations to obtain firstly  the vector $t_n = R_{n-1} \phi_n$ and secondly the outer vector product $t_n (\phi_n')^\top$. Finally, the update is $R_n = R_{n-1} - t_n (\phi_n')^\top$ which requires a final $O(d^2)$ addition operation of two matrices which is typically cheaper than $O(d^2)$ multiplication.  Therefore, the overall complexity is $O(d^2)$. Following reviewer's suggestion, we will include this analysis to the Appendix A.4.
>
> Regarding the theoretical justification and convergence analysis mentioned by the reviewer,  we would like to point out that our current work focuses on the methodological and computational/algorithmic development. Theoretical analysis requires substantial more work, and we plan to address it in the future.
>
> Regarding points 2 and 3 we agree with the reviewer that the preconditioner we propose is best suited for log-concave or close to log-concave distributions. For multi-modal distributions, as the mixture example mentioned by the reviewer, our precondioner could be much less effective (especially in case we have to average across modes of possibly "conflicting" curvature). To deal with such cases, our method needs to be extended, by following e.g. the suggestions from the literature given by the reviewer.  We can discuss this in our final section 7.
>
> In the discussion (Section 7) we mentioned the potential application of our method to HMC, motivated also by the wide popularity of HMC. One common intuitive argument in the literature (see e.g. Neal 2011) in favour of HMC compared to MALA is that HMC often can avoid random walk behaviour due to the use of the momentum. However, an interesting question is whether this argument still holds when an effective preconditioning is used for both MALA and HMC, or in such case any benefit from using HMC is reduced. We don't know the answer! But we believe that this is an interesting question to investigate in the future both experimentally and theoretically.

---

> > ### Comment · Reviewer_K8aZ · 2023-08-13
> > **Reply to authors**
> >
> > Do you have any comments and discussions on the references?

---

> > > ### Author Response · Authors · 2023-08-21
> > > **further comments about the references**
> > >
> > > Thanks for the discussion.
> > >
> > > Regarding the work of "Alfredo Garbuno-Inigo, et al. Interacting Langevin Diffusions: Gradient Structure And Ensemble Kalman Sampler, SIAM Journal on applied dynamical sysetm, 2019" it applies an ensemble of parallel Langevin diffusions  that shares the current ensemble covariance matrix as a preconditioner.  The authors study also the mean field limit, i.e when the number of  particles goes to infinity.
> > > We believe that such parallel-computing based approach could be combined with an ensemble precondtioner based on our inverse Fisher matrix. But we feel that this is a direction for future work, that it could be discussed for example in our final discussion section.  Note that in our current paper there is a single chain and not parallel multiple chains.
> > >
> > > Regarding the work of  Yifei Wang, et.al. Projected Wasserstein gradient descent for high-dimensional Bayesian inference, SIAM/ASA Journal on Uncertainty Quantification, 2022 and Yifei Wang, et.al. Accelerated Information Gradient Flow, Journal of Scientific computing, 2021 both use Wasserstein gradient flows. Note that in our work we use the standard discretized Langevin sampler (together with Metropolis-Hastings correction to ensure convergence to the target) and not a Wasserstein gradient descent approach. We believe that a possible direction for future work could be to use our algorithm in order to adapt the projection matrix $P_r$ in  Yifei Wang, et.al. Projected Wasserstein gradient descent for high-dimensional Bayesian inference.
> > >
> > > If the reviewer has further suggestions about how our work connects or it can be combined (in some future work) with the above references, we would like to hear about. Finally, we would like to thank again the reviewer for bringing these works into our attention.

---

### Official Review · Reviewer_NzaV · 2023-07-04

**Soundness:** 3 good
**Presentation:** 3 good
**Contribution:** 3 good
**Rating:** 7
**Confidence:** 4

**Summary:**

Preconditioned Metropolis-adjusted Langevin Algorithm is considered. Optimality of the preconditioning matrix, with respect to expected squared jump distance, is found to be proportional to an object related to the inverse Fisher information. In contrast to the Fisher, the expectation is taken with respect to model parameters, giving an object that is not position dependent (in contrast to related Riemannian manifold MALA, and RMHMC). An online algorithm, based on classical tools from Kalman filtering, is given for efficiently computing the inverse of a Monte Carlo approximation to the Fisher Information, leading to a novel adaptive preconditioned MALA method. Debiasing in the transient phase is discussed, and a Rao-Blackwellization scheme is given to reduce variance of the estimator. Experiments are conducted on a range of standard testbed tasks.

**Strengths:**

This paper provides a well principled approach to preconditioning MALA that shows improvements over both adaptive and Riemannian MCMC methods. In particular, it provides a quadratic approach to a Fisher-informed preconditioner that does not require cubic inversion/decomposition at every step, instead only needing quadratic updates. This approach outperforms the more expensive mMALA on nontrivial tasks.

The paper is communicated in an accessible way, and the experimental section covers a reasonable range of problems for samplers of this type.

**Weaknesses:**

The main body of the text claims that the Rao-Blackwellized version is significantly better, but this is not reflected in the numerics in Appendix E. This is not a weakness of the method itself, but appears misleading. When appendix E is mentioned, maybe "detailed comparison between the proposed methods" is clearer than "detailed results".

There are numerous grammatical errors and awkward phrasings, and the document would benefit from another read through with this in mind.

**Questions:**

Numerical performance of the non-centred version of FisherMALA are not reported, yet there is a burn-in period before the Fisher is computed that is usually used to mitigate the influence of the transient phase. How much effect does centering the scores have on performance, and how do the empirical means change throughout sampling?

The RB vs no-RB versions of your method appear to have performed very similarly, however with an acceptance rate tuned to <0.6, it seems natural to ask - how do they compare in terms of ESS/s? If the quadratic step is only being computed less than 60% of the time, does this negate the perceived/marginal benefits of using RB? This may also help demonstrate your performance advantage over mMALA.

**Limitations:**

Limitations are addressed in weaknesses/questions. There is little view for negative societal impact in this type of work.

---

> ### Author Rebuttal · Authors · 2023-08-09
>
> Regarding the Rao-Blackwellized version the reviewer is correct. Our numerical results in the Appendix E indeed show that the Rao-Blackwellized version is not significantly different than the non Rao-Blackwellized version, and both methods work well. Thank you for spotting this. Therefore, following reviewer's suggestion we will change the title of Section 4.1 from "Adapting to Rao-Blackwellized score function increments"to  "Adapting to score function increments" and remove or de-emphasise any claim about Rao-Blackwellization being significantly better.
>
> Regarding "detailed comparison between the proposed methods" is clearer than "detailed results". Thank you, we will add this change
> and resolve also the grammatical errors.
>
> Regarding the comment about "Numerical performance of the non-centred version of FisherMALA" we include in the rebuttal pdf file the Table 1 that includes also the "FisherMALA (non-centered)" version as requested by the reviewer. Note that due to space limitations in the Table we include only 5 benchmarks, i.e. the four from the main paper plus the Ripley logistic regression example.  Recall, that as mentioned in the main paper, the burn-in period before the Fisher is computed is 500 iterations in all runs. From this table we can see that "FisherMALA (non-centered)" performs worse than the other FisherMALA variants, and only on Ripley dataset works equally well with the rest. Of course, if we increase the 500-iterations burn-in window before Fisher computation starts, so that we get closer to convergence, "FisherMALA (non-centered)" could improve.  However, the main proposed FisherMALA versions that learn from the increments are much more robust to this choice and learn efficiently even when we start the adaptation early in the transient phase. We believe this robustness can be important in practice.
>
> For many targets (especially the high-dimensional ones) the empirical means of the gradients can be very far away from zero in the transient phase (due to space limitation we do not include a plot here in the rebuttal pdf).
>
> Regarding "RB vs no-RB versions running times", we would like to point to an answer given above to reviewer bdsU, where we provided a Table of actual running times (experiments were done by using a v100 GPU) in the most expensive MNIST benchmark. Note that all MALA variants have similar running times (besides mMALA) and this is because the gradient computation of the log target dominates the running time and the quadratic cost needed for FisherMALA is somehow insignificant compared to the gradient computation. mMALA is slower because it requires computation of the Hessian plus its $O(m^3)$ matrix decomposition which adds an extra cost, while the slowest is HMC because  it requites multiple gradient evaluations per iteration. FisherMALA-no-RB is not included in the table but it is like FisherMALA (i.e. recall that FisherMALA in the main paper corresponds to the variant FisherMALA-with-RB as denoted in Table 7 in the Appendix).

---

> > ### Comment · Reviewer_NzaV · 2023-08-18
> > **Response**
> >
> > I'm happy with this response and have raised my score accordingly.

---

### Official Review · Reviewer_bdsU · 2023-07-07

**Soundness:** 4 excellent
**Presentation:** 4 excellent
**Contribution:** 4 excellent
**Rating:** 8
**Confidence:** 4

**Summary:**

This is a nice paper that develops a preconditioner for improved sampling of complex target distributions using gradient history.  The algorithm is well presented and is placed in context of other recent proposals from the literature.   There are several numerical experiments presented on small to medium sized examples that support the idea.

A nice feature is that the paper is very readable which will enhance the uptake of the idea.

The numerical experimentation is good but not exhaustive, nonetheless convincing to this reader.

**Strengths:**

Reasonable overview of literature.   Careful presentation.  Readable style.

**Weaknesses:**

Limited experimentation in high dimensional settings.     Limited ablation study.



**Questions:**

What were wall clock times for all the different computations?  In particular they should be reported for the larger models.

You give many results in the table for Bayesian logistic regression.  How did you parameterize the various methods?

How does the performance of the method presented in the paper scale up in a parallel computing setting?


**Limitations:**

The authors miss the chance to introduce the work by indicating the importance of sampling in large scale data science in providing improved robustness and the potential for uncertainty quantification.  I would suggest that the introduction be modified to develop this theme.   By its very nature, this work has potential to strengthen the foundations of AI and this should at least be mentioned.

---

> ### Author Rebuttal · Authors · 2023-08-09
>
>  "What were wall clock times for all the different computations? In particular they should be reported for the larger models."
>
> In the Table below we report the overall running time for all samplers in the largest MNIST Bayesian logistic regression problem. These running times were obtained by running all algorithms using a v100 GPU. We can observe the following. All MALA variants have similar running times (besides mMALA) and this is because the gradient computation of the log target dominates the running time and the quadratic cost needed for FisherMALA is somehow insignificant compared to the gradient computation. mMALA is slower because it requires computation of the Hessian plus the Hessian matrix decomposition which adds an extra cost, while the slowest is HMC because it requites multiple gradient evaluations per iteration.
>
> **Running times (in seconds) for MNIST**
>
> **MALA**          $1900.086 \pm 60.186$
>
> **AdaMALA**    $1927.065 \pm 190.311$
>
> **HMC**            $8046.461 \pm 272.085$
>
> **mMALA**        $4442.111 \pm 207.497$
>
> **FisherMALA**  $1910.230 \pm 112.168$
>
> "You give many results in the table for Bayesian logistic regression. How did you parameterize the various methods?"
>
> Section 5.1 provides details about how adaptation is done for all the different samplers and for all targets (including all Bayesian logistic regression examples in the Appendix, i.e. all runs follow Section 5.1), such as how the global $\sigma^2$ is adapted, number of burn-in iterations etc. Please let us know if this information is not sufficient.
>
> "How does the performance of the method presented in the paper scale up in a parallel computing setting?"
>
> If we use multiple parallel MCMC chains then we would like perhaps these chains to communicate information and adapt a preconditioner that uses gradients from all chains. How to do this efficiently is an interesting research topic, but we believe that this is beyond the scope of the current work and it is suitable for future research.
>
> "The authors miss the chance to introduce the work by indicating the importance of sampling in large scale... "
>
> Thank you for this comment. We will try to follow reviewer's suggestion and add a related sentence in the first paragraph of the introduction.

---

### Author Rebuttal · Authors · 2023-08-09

We thank all reviewer for the time and for their constructive comments.  Responses to comments of each individual reviewer are given after each corresponding review.

Here, in the general comments we would like to point out to all reviewers that after the submission we added a correction to Section 3 of the paper since we noticed  that the optimal preconditioner must be the minimum and not the maximum of the expected squared jumped distance (ESJD).  And indeed the inverse Fisher matrix that we propose in the paper is the minimum of the ESJD. Here, we provide the part of Section 3 that has been re-written.  Nothing else changes in the rest of the paper apart from the above change, ie.  that the optimal preconditioner which is the inverse Fisher, is the minimum and not the maximum of ESJD. For completeness here we provide the text that has been re-written in Section 3 and also the proof that the inverse Fisher is the minimizer of ESJD.

**The main changes in Section 3 are the following**

We develop a method for selecting the preconditioning through the optimization of an objective function.  This method uses the observation that an effective preconditioning should make the target more well-conditioned or symmetrized which subsequently allows to use the largest possible discretization step size $\sigma^2$ in MALA.  An illustration of this is shown in Figure 1 in the attached pdf of this rebuttal
where our proposed inverse Fisher preconditiorer indeed gives the largest $\sigma^2$ when compared to other MALA samplers.

Given the time discretized Langevin diffusion
$$
x_{t + \delta} - x_t = \frac{\delta}{2} A \nabla \log \pi(x_t)  + \sqrt{A} (B_{t+ \delta} - B_t),  \ \ \text{where} \ \  B_{t+ \delta} - B_t \sim \mathcal{N}(0,\delta I).
$$
the expected squared jumped distance is $J(\delta, A) = E[||x_{t+\delta}  - x_t ||^2]$ where  $x_t \sim \pi$.   To control discretization error we can impose an upper bound constraint $J(\delta, A) \leq \epsilon$ for small $\epsilon > 0$. To obtain a preconditioning $A$ that "symmetrizes" the target as much as possible we would like to maximize the discretization step size $\delta$ subject to $J(\delta, A) \leq \epsilon$. Since $J(\delta, A)$ monotonically increases with $\delta$, the maximum $\delta^*$ satisfies $\min_A J(\delta^*, A) = \epsilon$. This means that the optimal preconditioning $A^*$ is obtained by minimizing the expected squared jumped distance, i.e.  by solving $\min_A J(\delta, A)$ under some global scale constraint on $A$, as stated next.


**Proposition 3**

Suppose $A$ is a symmetric positive definite matrix satisfying
$\text{tr}(A) = c$, with $c>0$ a constant.  Then the objective $J(\delta, A)$ is miminized for $A^*$ given by
$$
A^* = k \mathcal{I}^{-1}, \ \ k=\frac{c}{\sum_{i=1}^d \frac{1}{\mu_i} }, \ \ \ \mathcal{I} = E_{\pi(x)}
\left[ \nabla \log \pi(x)
\nabla \log \pi(x)^\top \right],
$$
where $\mu_i$s are the eigenvalues of $\mathcal{I}$ assumed to satisfy $0 < \mu_i < \infty$.



**For completeness also we provide here in the rebuttal the new proof of Proposition 3**

**Proof of Proposition 3**

The expected squared jumped distance is written as
$$
J(\delta, A) = \text{tr} \left(\frac{\delta^2}{4} A \mathcal{I} A +  \delta A \right)
=\frac{\delta^2}{4}
\text{tr}(A \mathcal{I} A) + \delta c,
$$
where we used
$\text{tr}(A) = c$. Since $c$ is  a constant to minimize $J(\delta, A)$ is the same as minimizing $\text{tr}(A \mathcal{I} A)$ under the constraint that $A$ is symmetric positive definite matrix and $\text{tr}(A)=c$.  To deal with the equality constraint  we consider the Lagrangian
$
\text{tr}(A \mathcal{I} A) - \lambda ( \text{tr}(A) - c).
$
By taking derivatives wrt $A$ (using the matrix derivative identities  $\frac{\partial}{\partial X} \text{tr}(X B X) = X^\top B^\top + B^\top X^\top$ and $\frac{\partial}{\partial X} \text{tr}(X) = I_d$ for arbitrary $d \times d$ square matrices $X, B$) and setting to zero we obtain the linear equation
$
A^\top \mathcal{I} + \mathcal{I} A^\top =  \lambda I_d,
$
where we used that $\mathcal{I}$ is a symmetric matrix.  This is a set of linear equations
and given that each eigenvalue $\mu_i$ of $\mathcal{I}$ satisfies $0 < \mu_i < \infty$,  so that $\mathcal{I}$ is invertible,  there is an unique solution given by $A = (1/2) \lambda \mathcal{I}^{-1}$. The Lagrange multiplier $\lambda$  is chosen so that $\text{tr}(A)=c$ which leads to the optimal $A^*$
$$
A^* = \frac{c}{\sum_{i=1}^d \frac{1}{\mu_i}} \mathcal{I}^{-1}.
$$
$A^*$ turned out to be symmetric and  positive definite as  desired.  For this $A^*$ the optimal loss value is $\text{tr}(A^* \mathcal{I} A^*) = \frac{c^2}{\sum_{i=1}^2  \frac{1}{\mu_i}}$, for which we further need to disambiguate whether this is the global minimum or maximum. We can do this by choosing a different matrix that satisfies the constraint $\text{tr}(A)=c$ and  compare its loss with the optimal loss  $\frac{c^2}{\sum_{i=1}^d  \frac{1}{\mu_i}}$ . For example, one such matrix is $A = \frac{c}{d} I_d$, which has loss value $\frac{c^2  (\sum_{i=1}^d \mu_i)}{d^2}$. Then by using the Cauchy-Schwarz inequality $d^2 = (\sum_{i=1}^d \frac{\sqrt{\mu_i}}{\sqrt{\mu_i}})^2 \leq (\sum_{i=1}^d \mu_i ) (\sum_{i=1}^d \frac{1}{\mu_i})$
we obtain $\frac{c^2  (\sum_{i=1}^d \mu_i)}{d^2} \geq  \frac{c^2  (\sum_{i=1}^d \mu_i)}{  (\sum_{i=1}^d \mu_i ) (\sum_{i=1}^d \frac{1}{\mu_i})}$ = $ \frac{c^2}{\sum_{i=1}^d  \frac{1}{\mu_i}}$.
This shows that $A^*$ achieves the global minimum which completes the proof.

---

### Decision · Program_Chairs · 2023-09-21

**Decision:**

Accept (poster)

**Comment:**

Reviewers for this paper were quite positive. I tend to side with the authors regarding some of the reviewer’s concerns about a lack of theoretical justification; I see this paper as a methods/algorithms paper and I don’t think a thorough theoretical analysis is necessary for acceptance. I encourage the authors to incorporate their responses from the rebuttal into the paper, in addition to making an editing pass throughout the paper focusing on grammar and typos. Overall, I would like to congratulate the authors on their great work.